# Chromatin remodeling in bovine embryos indicates species-specific regulation of genome activation

Michelle M. Halstead[1], Xin Ma[1,2], Chuan Zhou[1], Richard M. Schultz[3,4] & Pablo J. Ross [1✉]

The shift from maternal to embryonic control is a critical developmental milestone in pre-implantation development. Widespread transcriptomic and epigenetic remodeling facilitate this transition from terminally differentiated gametes to totipotent blastomeres, but the identity of transcription factors (TF) and genomic elements regulating embryonic genome activation (EGA) are poorly defined. The timing of EGA is species-specific, e.g., the timing of murine and human EGA differ significantly. To deepen our understanding of mammalian EGA, here we profile changes in open chromatin during bovine preimplantation development. Before EGA, open chromatin is enriched for maternal TF binding, similar to that observed in humans and mice. During EGA, homeobox factor binding becomes more prevalent and requires embryonic transcription. A cross-species comparison of open chromatin during preimplantation development reveals strong similarity in the regulatory circuitry underlying bovine and human EGA compared to mouse. Moreover, TFs associated with murine EGA are not enriched in cattle or humans, indicating that cattle may be a more informative model for human preimplantation development than mice.

[1] Department of Animal Science, University of California, Davis, CA 95616, USA. [2] College of Animal Science and Technology, Jilin Agricultural University, Changchun, Jilin Province 130118, China. [3] Department of Anatomy, Physiology and Cell Biology, School of Veterinary Medicine, University of California, Davis, CA 95616, USA. [4] Department of Biology, University of Pennsylvania, Philadelphia, PA 19104, USA. ✉email: pross@ucdavis.edu

Preimplantation development encompasses several critical milestones as embryos progress from fertilization to implantation. Fusion of the transcriptionally quiescent oocyte and sperm results in a zygote with two haploid pronuclei. Gametic syngamy is completed after the first round of DNA replication, when pronuclear membranes dissolve, allowing maternal and paternal chromosomes to intermingle on the metaphase plate. Subsequent rounds of cleavage ultimately lead to formation of a blastocyst. However, the cleavage-stage embryo must first complete the maternal-to-zygotic transition (MZT), wherein the embryo assumes control over its own continued development by degrading oocyte-derived products and initiating its own transcriptional program. This dramatic change in gene expression proceeds gradually; minor embryonic genome activation (EGA) results in low levels of transcription in early cleavage-stage embryos[1,2], and leads to major EGA, which involves widespread transcription of embryonic genes[3]. This shift from maternal dependence to self-sufficiency serves at least three functions: elimination of oocyte-specific messages, replenishment of transcripts that are common to both the oocyte and the embryo, and generation of novel embryonic-specific transcripts[4].

While embryos undergo this massive shift in transcriptional program during the MZT, wide-spread changes in chromatin structure[5] and epigenetic information are simultaneously restructuring the embryonic epigenome. Chromatin remodeling is necessary to erase gamete-specific signatures and establish an open chromatin landscape that supports embryonic transcriptional programs. Specifically, chromatin structure defines the genomic context within which transcriptional machinery can operate, thereby determining the cell-specific gene expression patterns that confer cell identity and function.

Following fertilization, the zygotic genome is globally demethylated[6], and this loss of DNA methylation coincides with global decreases in repressive histone modifications[5,7]. Overall, epigenetic factors linked to relaxed chromatin are more abundant in mouse zygotes, whereas factors implicated in chromatin compaction become more prevalent during EGA[8], pointing to a more permissive chromatin landscape in pre-EGA embryos. Indeed, mouse zygotes present increased histone mobility[9], highly dispersed chromatin[10], and lack chromocenters (congregations of pericentromeric heterochromatin)[11,12]. Moreover, in mouse, 3-D chromatin architecture is largely absent after fertilization, and is gradually established throughout preimplantation development, which facilitates long-distance chromatin interactions in later stage embryos[13], and indicates increasing chromatin compaction and organization. The co-occurrence of genome activation and chromatin remodeling raises an interesting causality dilemma, namely, whether chromatin remodeling is necessary for transcription activation or whether transcription activation leads to chromatin remodeling. Several maternal products prompt transcription initiation by altering chromatin structure[14–16], and some chromatin compaction occurs in the absence of embryonic transcription[13]. However, inhibiting embryonic transcription pervasively disrupts establishment of open chromatin during EGA[17,18], suggesting that EGA and chromatin remodeling are likely interdependent.

In mouse and human preimplantation embryos, enrichment for chromatin accessible sites is gradually established as development progresses[17–22]; however, these accessible sites demonstrate different motif enrichment patterns, implicating distinct sets of transcription factors (TFs) in either murine (RARG, NR5A2, ESRRB)[17], or human EGA (OTX2, GSC, POU5F1)[18,20]. Although some TFs appear to regulate EGA in multiple species, i.e., Krüppel-like factors (KLFs)[17,18,23], double homeobox (DUX)[24–27], ZSCAN4[28,29], and CTCF[17,18], the level of conservation across mammals remains unclear. In fact, the timing of genome activation is highly species-specific. RNA-seq studies indicate that the major wave of EGA in mice occurs during the 2-cell stage[30], in humans during the 4-to-8-cell stage[31], and in cattle during the 4-to-8-cell stage[32]. The relative timing at which mice activate their genomes could indicate that the mechanism behind murine EGA differs significantly from other species', which would have significant implications for modeling human pre-implantation development. In particular, the timing of bovine EGA more closely resembles that of human EGA, as do global changes to histone post-translational modifications. For instance, the active mark trimethylation of lysine 4 on histone 3 (H3K4me3) decreases in global abundance during human[33] and bovine EGA[34], but increases during murine EGA[35,36]. However, the chromatin remodeling events that underscore bovine pre-implantation development have yet to be cataloged, and it remains unclear whether the regulation and execution of the MZT in cattle resembles what has been observed in humans.

Here we report progressive gains in ATAC-seq peaks throughout bovine preimplantation development. Peaks established prior to major EGA are predominantly enriched for binding sites of maternally provided factors (CTCF, NFY, SP factors, KLFs), whereas homeobox TF binding (OTX1, GSC, DUX factors) becomes more prevalent during major EGA, and depends on embryonic transcription. Comparison of the open chromatin landscape during preimplantation development in cattle, human, and mice reveals that the regulatory circuitry underlying bovine and human EGA is very similar and distinct from mouse. Moreover, TFs that appear to regulate murine EGA (RARG, NR5A2, ESRRB) demonstrate no enrichment in cattle or humans, suggesting that cattle are a more informative model for human preimplantation development than mice.

## Results

**Dynamics of open chromatin in bovine preimplantation embryos.** For each developmental stage, ATAC-seq libraries were prepared from pools of cells derived from three separate oocyte collections. A subset of embryos from each collection was also cultured in the presence of the transcriptional inhibitor, α-amanitin, to interrogate the relationship between embryonic transcription and chromatin remodeling (Fig. 1a). Between 30 and 87 million nonmitochondrial monoclonal uniquely mapped reads were collectively obtained for each developmental stage and cell type: germinal vesicle-stage (GV) oocytes, 2-, 4-, 8-cell embryos, morula, inner cell masses (ICM), and embryonic stem cells (ESC)[37] (Supplementary Table 1). Of note, because pools of embryos were used as input for ATAC-seq, the resulting chromatin accessibility profiles are representative of cell populations with some level of heterogeneity, especially for later stages of development. Genome-wide ATAC-seq coverage demonstrated a striking shift in the open chromatin landscape between 4- and 8-cell embryos (Fig. 1b), suggesting that large-scale chromatin remodeling coincides with major EGA (Supplementary Fig. 1). Strong correlation of genome-wide signal between biological replicates (Supplementary Table 2) indicated that both the technique and embryo production were robust, generating comparable chromatin accessibility profiles across different rounds of oocyte collection and embryo production (Fig. 1c). Reads from replicates were pooled together to obtain greater sequencing depth and maximize power for identifying regions of open chromatin. To gauge changes in chromatin accessibility throughout development, regions with enriched signal, or peaks, were called for each stage of development. Pooled reads from replicates were randomly subsampled to standard depths before calling peaks to minimize bias from sequencing depth. Peaks were called from either 30 million monoclonal uniquely mapped reads when comparing control embryos at different developmental

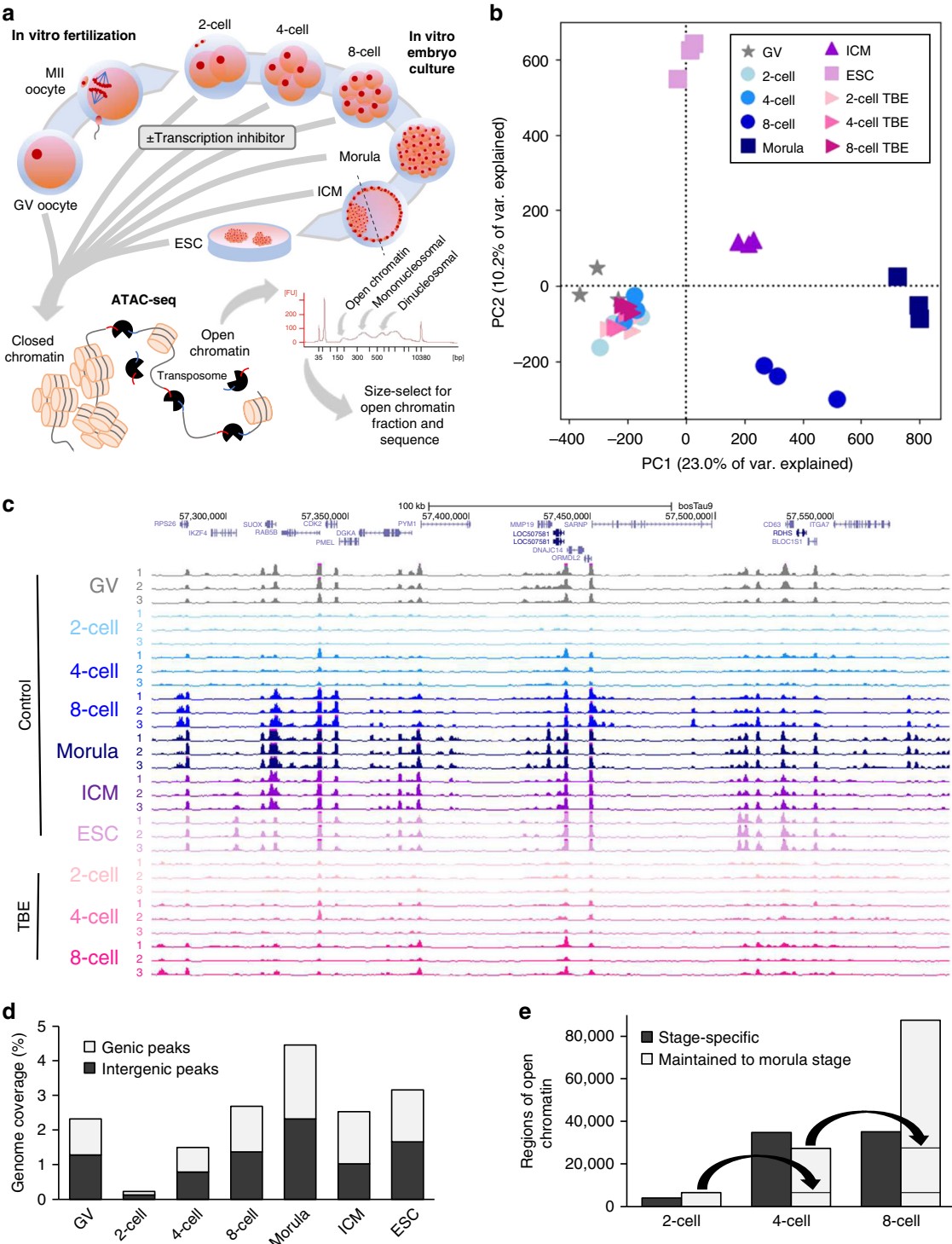

**Fig. 1 Chromatin accessibility in bovine oocytes and in vitro preimplantation embryos. a** Schematic of in vitro embryo production and ATAC-seq library preparation. **b** Principal components analysis (PCA) of ATAC-seq signal, normalized by fragments per kilobase million (FPKM), in 500 bp windows covering the whole genome. **c** Normalized coverage (FPKM) of replicate ATAC-seq libraries for each stage of development. **d** Proportion of genome that is covered by genic and intergenic ATAC-seq peaks, called from 30 million reads for each developmental stage. **e** Categorization of ATAC-seq peaks in 2-, 4-, and 8-cell embryos into stage-specific and maintained peaks (accessibility maintained up until the morula stage). Maintained peaks are carried over from latter stages to show cumulative maintained peaks. Source data are provided as a Source Data file.

stages, or 20 million reads when comparing transcription blocked embryos (TBE) with control embryos. Iterative random sub-sampling resulted in consistent ATAC-seq peak sets, indicating that subsampling did not introduce additional variation to peak calling (Supplementary Fig. 2).

As observed in humans and mice[17,18,20–22], ATAC-seq peaks were progressively gained throughout bovine preimplantation development, with the lowest enrichment for accessible sites in 2-cell embryos (Fig. 1d). Considering that multiple studies in pre-EGA embryos indicate highly dispersed chromatin[9,10], lack of

chromocenters[11,12], and lack of higher order chromatin organization[13], this decreased enrichment for open chromatin in 2-cell embryos could reflect a highly permissive chromatin structure, resulting in similar levels of accessibility across the genome and lack of specific peak enrichment. Because assays that employ endonucleases—e.g., ATAC-seq and DNase-seq—depend on enrichment for cutting events at consistently accessible loci, genome-wide chromatin relaxation would lead to random cutting events, resulting in the observed high background and low enrichment in 2-cell embryos. Attempts to use endonuclease-based methods to profile open chromatin in mouse zygotes[17,22] and human 2-cell embryos[20,21] have encountered similar difficulties with low enrichment. On the other hand, a recent study that profiled chromatin accessibility with a technique that was not based on endonuclease activity found that genome-wide accessibility in human embryos actually decreased from the zygote stage onward[19]. Collectively, these data indicate that global chromatin relaxation is a shared characteristic of human, bovine, and murine pre-EGA embryos. It is tempting to speculate that this naïve chromatin structure acts as an epigenetic "blank slate", which is then gradually compacted and structured to meet the needs of the developing embryo.

As chromatin becomes more structured, specific loci become more consistently accessible, and can thus be detected as enriched areas of open chromatin by ATAC-seq. Indeed, ATAC-seq peaks were progressively gained in 4-cell, 8-cell, and morula stage embryos (Fig. 1d). Intriguingly, many peaks were only apparent at a specific stage, whereas other peaks were maintained throughout later stages (Fig. 1e), suggesting that chromatin remodeling serves two functions: progressive establishment of a chromatin landscape that will support further development, and transient stage-specific regulation.

**Stage-specific open chromatin has distinct functionality.** To delve further into the potential function of ATAC-seq peaks that were gained or lost between consecutive developmental stages (Fig. 2a), computational footprinting was used to identify regions that were actively bound by TFs at each stage. These footprints were matched to known binding motifs[38], and TF binding activities were compared between consecutive stages (Supplementary Fig. 3, Supplementary Data 1–7). Both motif enrichment and binding activity changed dramatically as development progressed, demonstrating that chromatin remodeling subjects each stage of development to a distinct regulatory circuitry (Fig. 2b). Sequence enrichment within intergenic ATAC-seq peaks revealed that the binding motifs of distinct groups of TFs gained accessibility at each developmental stage (Supplementary Table 3). These enriched sequences corresponded to the known binding motifs of several TFs that were either maternally provided or embryonically expressed (Supplementary Fig. 4).

Over 30,000 ATAC-seq peaks were unique to GV oocytes (Fig. 2a), and those which were intergenic ($n = 20,959$) were notably enriched for the binding motifs of RFX factors and folliculogenesis specific bHLH factor (FIGLA) (Fig. 2b, Supplementary Fig. 5). Although very few ATAC-seq peaks were detected at the 2-cell stage, nearly 70,000 peaks were gained during the 2- to 4-cell transition. Surprisingly, nearly half of these peaks were only transiently accessible (Fig. 1e). Four-cell-specific open chromatin was most significantly enriched for binding motifs of NFkB family members (Fig. 2c), which was particularly intriguing because NFkB activation is required in mouse 1-cell embryos for development to progress past EGA[39]. Although NFkB factors are maternally provided (Fig. 2d), they are initially sequestered in the cytoplasm until they translocate into the nucleus at the early 1-cell stage in mice[39] and the 4-to-8 cell stage

in cattle[40]. In particular, RELA – one of the NFkB subunits capable of activating expression – binds DNA with increased frequency in cattle embryos compared to oocytes[40], suggesting that NFkB activation of target gene expression could be one of the first events in a cascade leading to major EGA. In fact, one of the few genes with upregulated expression in 4-cell embryos, as compared to MII oocytes, was *TRIM8*, a positive regulator of NFkB activity (Supplementary Fig. 6). Overall, NFkB footprints marked promoters of genes involved in negative regulation of cell differentiation at the 4-cell stage (Supplementary Table 4, Supplementary Data 8), including several potential regulators of EGA that were substantially upregulated between the 4- and 8-cell stages (Supplementary Fig. 7). Nevertheless, the contribution of NFkB activity to minor EGA in cattle has yet to be established, but certainly warrants further research.

In contrast to open chromatin in pre-EGA embryos, ATAC-seq peaks that were gained at the 8-cell stage were predominantly enriched for binding motifs of homeobox TFs, including OTX2, GSC, CRX, PHOX2A, PAX7, and PITX1 (Fig. 2e), although only *OTX1*, *OTX2*, and *PITX2* were appreciably expressed during the 8-cell stage (Fig. 2f). Based on the presence of OTX and PITX footprints in promoters, these homeobox TFs appear to target genes involved in negative regulation of cell differentiation (Supplementary Table 4, Supplementary Data 8). De novo motif enrichment revealed that intergenic ATAC-seq peaks that were gained at the 8-cell stage were most significantly enriched for a motif that corresponded to the DUX family, which has been extensively implicated in EGA regulation[24–27]. More than a third of the accessible intergenic loci in 8-cell embryos harbored a sequence that corresponded to DUX binding motifs (Fig. 2g). DUX factors are increasingly recognized as potential developmental pioneer factors, participating in chromatin remodeling[25,27], and expression of repetitive elements and cleavage-stage genes[24–26]. In cattle, *DUXA* is expressed transiently and strongly during the 8- and 16-cell stages (Fig. 2h), similar to *DUX4* and *Dux* expression in human and mouse embryos, respectively[25,26]. Although *Dux* expression in mice can be induced by the maternal factors DPPA2 and DPPA4[41], only *DPPA3* is maternally provided in cattle (Supplementary Fig. 8). Knockdown of *DPPA3* impairs the developmental competency of mouse and bovine embryos[42,43], highlighting its potential as a key regulator. Although the maternal factors that induce *DUXA* expression in cattle remain unclear, the synchronized upregulation of *DUXA* and increased accessibility of its binding sites during bovine development suggest that DUXA may play an important role in bovine EGA. This hypothesis is especially intriguing, considering that DUX family members are implicated in murine and human EGA, and are highly conserved and specific to placental mammals[44]. Further functional validation will be necessary to determine if DUXA is required for bovine embryogenesis, or if is an important but nonessential regulator of EGA, as in mice[45–47].

The expression profile of *DUXA* in bovine preimplantation development closely mirrored that of *ZSCAN4*, another TF implicated in EGA (Supplementary Fig. 9)[48]. *ZSCAN4* is a downstream target of DUX factors in humans[25] and mice[24,26], and the coordinated expression of these two factors in cattle certainly suggests that a similar mechanism may be at play. ZSCAN4 depletion results in developmental arrest at the time of EGA in mice[28] and cattle[29], supporting an important role for this TF. A recent report indicates that ZSCAN4 binds microsatellites in mouse embryos[49]. These binding sites would be problematic to identify in our analysis, given the difficulty of aligning short reads to repetitive sequences. Nevertheless, ZSCAN4 footprints were apparent in the promoters of 544 genes at the 8-cell stage, and these genes were functionally enriched for negative regulation of

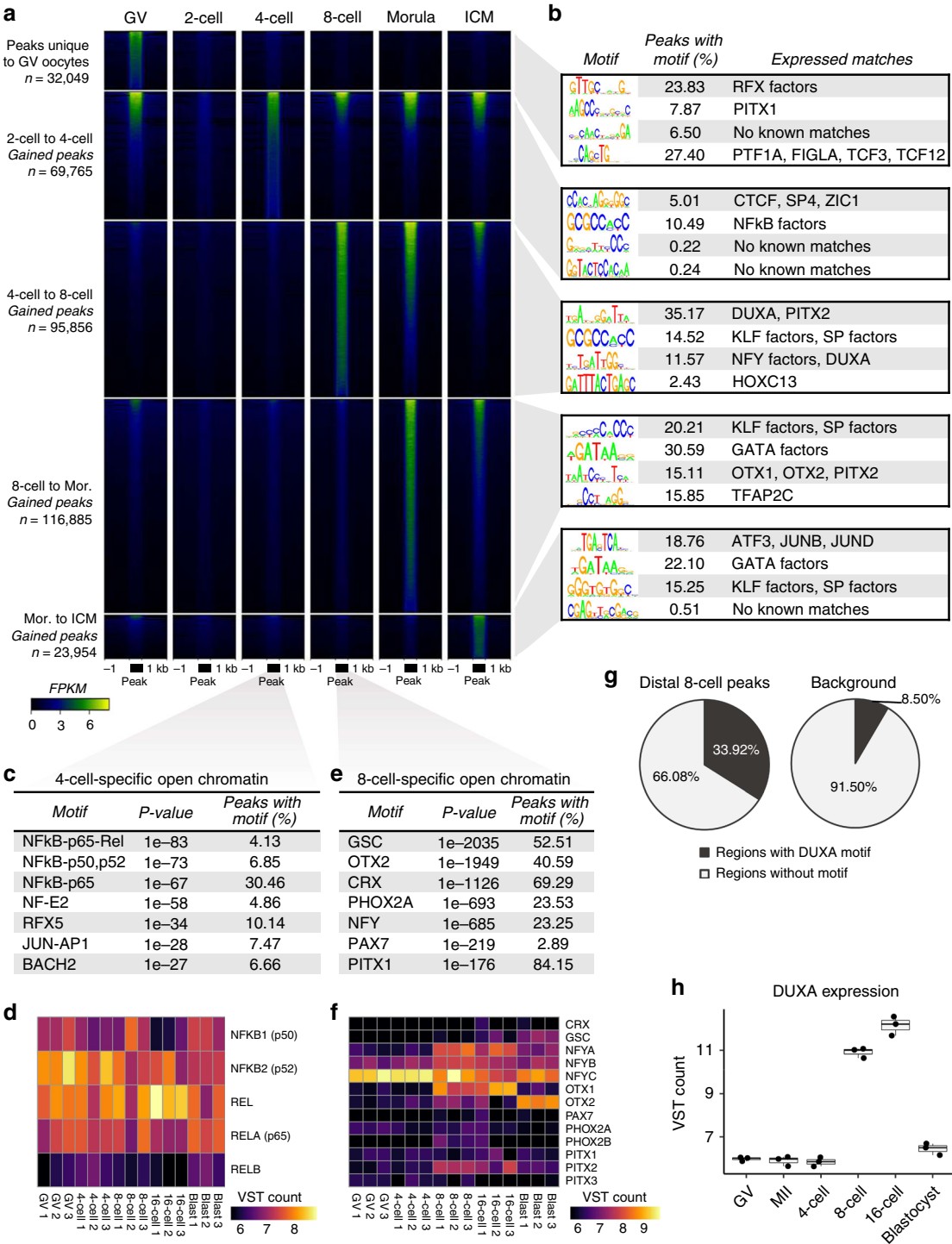

**Fig. 2 Gradual establishment of open chromatin enriched in regulatory motifs. a** Normalized ATAC-seq signal (FPKM) shown at peaks that were either unique to GV oocytes, or which were gained between consecutive developmental stages. Peaks at each stage were called from 30 million reads, and were scaled to 500 bp. **b** Top four de novo motifs enriched in intergenic peaks, matched to known motifs (log odds match score >0.6) of TFs that were expressed at the given stage. **c** Top seven known motifs enriched in 4-cell specific peaks. Binomial test *p* values reported. Adjustment for false discovery rate (FDR) yielded *Q* values less than 1e–4. **d** Variance stabilized transformed (VST) expression counts of TFs corresponding to enriched motifs in 4-cell specific open chromatin. **e** Top seven known motifs enriched in 8-cell specific peaks. Binomial test *p* values reported (all FDR adjusted *Q* values <1e–4). **f** VST expression counts of TFs corresponding to enriched motifs in 8-cell specific open chromatin. **g** Proportion of intergenic 8-cell peaks overlapping the de novo motif most closely matching the DUXA motif, relative to background. **h** VST expression counts of *DUXA* throughout development (*n* = 3 biologically independent samples). Boxplots indicate the median and interquartile range (IQR), and whiskers span 1.5 times the IQR. RNA-seq data from Graf et al.[32]. Source data are provided as a Source Data file.

cell differentiation and apoptosis, as well as positive regulation of proliferation (Supplementary Table 4, Supplementary Data 8).

Extensive chromatin remodeling continued to take place after major EGA. From the 8-cell to the morula stage, over 100,000 ATAC-seq peaks appeared (Fig. 2a), and those that were intergenic were primarily enriched for KLF, SP, and GATA binding motifs (Fig. 2b; Supplementary Table 3). Differential analysis of TF binding activity in 8-cell embryos and morula, based on TF footprinting, revealed significantly increased KLF and SP binding activity in morula, although GATA binding activity did not significantly increase (Supplementary Fig. 4). Nonetheless, almost 20,000 GATA footprints were evident in open chromatin at the morula stage and occurred in promoters of genes involved in negative regulation of differentiation (Supplementary Table 4, Supplementary Data 8). Genes marked by KLF and SP footprints in 8-cell embryos and morula demonstrated substantial enrichment for a variety of biological processes, including cell-cell adherens junctions (Supplementary Table 4, Supplementary Data 8).

Comparatively fewer ATAC-seq peaks were gained during the morula to ICM transition ($n = 23,954$) (Fig. 2a). Intergenic peaks gained in ICM were also enriched for GATA, KLF, and SP motifs, but, in contrast to ATAC-seq peaks gained in morula, were significantly enriched for JUN factor motifs (Fig. 2b). Footprinting analysis corroborated this enrichment, showing that JUN factors exhibited stronger binding in ICM than in morula (Supplementary Fig. 4). Increased JUN binding may reflect the beginning of primitive endoderm specification, as c-JUN induces expression of endoderm markers[50].

**Gradually established open chromatin sets the stage for EGA.** Although many ATAC-seq peaks were stage-specific, a stable open chromatin landscape was also progressively established after fertilization. As early as the 2-cell stage, ATAC-seq peaks appeared and were maintained post-EGA until at least the morula stage. Maintained ATAC-seq peaks were enriched for CTCF footprints (Fig. 3a; Supplementary Table 5), especially those that were first established during the 4-cell stage. CTCF binding delineates chromatin loop boundaries, thus determining the genomic space within which genes can interact with their regulatory elements[51]. Therefore, enrichment of CTCF binding in maintained peaks could indicate a gradual establishment of a stable 3-D chromatin architecture in preparation for major EGA. This interpretation is consistent with 3-D chromatin architecture in mouse embryos, which is greatly diminished after fertilization and then gradually established throughout preimplantation development, facilitating long-distance chromatin contacts, i.e., promoter-enhancer interactions, in later stage embryos[13]. Indeed, transcription during minor EGA in mouse is primarily driven by proximal promoters, whereas enhancers are dispensable for transcription until major EGA[52]. The global reorganization of 3-D chromatin architecture in the mouse has also been observed in bovine embryos: the nonradial establishment of chromosome territories, when gene-poor chromosomes tend to localize to the nuclear periphery more often than gene-rich regions, is first established at the 8-cell stage[53]. Collectively, these results suggest that gradual establishment of 3-D chromatin architecture may be a conserved feature of pre-EGA embryos, although the specific mechanisms and factors regulating this restructuring remain unknown.

Maintained peaks were also more enriched for KLF motifs than stage-specific peaks (Fig. 3b), and several *KLFs* were maternally provided (Fig. 3c), including *KLF4*, a master regulator of induced pluripotent stem cell reprogramming[54]. KLF4 contributes to reprogramming by mediating pluripotency-associated enhancer-promoter contacts[55]; thus, the concurrent establishment of open chromatin corresponding to CTCF and KLF binding motifs could indicate that pre-EGA embryos are priming a similar mechanism for use during major EGA, especially because about 50% of activated genes during bovine major EGA contain KLF motifs in their promoters[23].

In fact, ATAC-seq peaks that were maintained from the 2- and 4-cell stages onward occurred in genic regions more often than stage-specific peaks (Fig. 3d) and marked the promoters of genes that were functionally enriched for housekeeping functions (Fig. 3e), including 33 of the 51 genes that were upregulated in 4-cell embryos compared to MII oocytes (Supplementary Fig. 6). Moreover, maintained ATAC-seq peaks were strongly enriched for NFY and SP1 binding sites (Supplementary Table 5), which is highly reminiscent of chromatin remodeling in mouse, as proximal promoters enriched for NFY motifs are some of the first regions to gain accessibility[22]. Of note, NFY enhances binding of the pluripotency factors POU5F1, SOX2, and KLF4 to their recognition motifs[56], and is clearly involved in murine EGA, with NFY knockdown embryos presenting impaired open chromatin establishment and downregulation of gene expression[22]. Similarly, SP1 is essential for early mouse development, with knockout embryos arresting at day 11 of gestation[57]. Intriguingly, human zygotes and 4-cell embryos also demonstrate KLF, SP1, and NFY motif enrichment in open chromatin[21], overall suggesting that a conserved set of maternal regulators participate in chromatin remodeling and transcription activation in pre-EGA embryos.

**Maternal factors and transcription drive chromatin remodeling.** Considering that embryonic transcription is limited before the 8-cell stage in cattle, the appearance of ATAC-seq peaks in pre-EGA embryos suggests that maternal factors actively participate in chromatin remodeling. To further dissect the maternal contribution to epigenetic reprogramming and EGA, embryos were cultured in the presence of α-amanitin to inhibit POLR2-dependent transcription elongation. Treatment with α-amanitin prevents development of bovine embryos past the 8-to-16-cell stage[58]. Loss of embryonic transcription dramatically repressed the appearance of ATAC-seq peaks (Fig. 4a); 64% of ATAC-seq peaks that should have been gained at the 4-cell stage failed to appear without embryonic transcription, and in 8-cell TBEs, 96% of ATAC-seq peaks that should have appeared were not present (Fig. 4b), disrupting the chromatin state of key genes, such as *KLF4* (Fig. 4c).

Interestingly, transcription inhibition preferentially disrupted stage-specific ATAC-seq peaks, whereas maintained peaks that were established in 2- and 4-cell embryos appeared despite transcription inhibition (Fig. 4d). Transcription-independent maintained peaks marked the promoters of nearly 60% of embryonically-expressed genes ($n = 1,038/1,784$ genes identified from data from Bogliotti et al.[23]; Supplementary Fig. 10), suggesting that maternal factors actively remodel the local chromatin structure of target genes, possibly priming them for expression later in development. In fact, transcription inhibition did not affect the binding activity of maternal factors CTCF and KLFs (Supplementary Fig. 11), which were actively bound in 2- and 4-cell maintained open chromatin (Fig. 3a, b). On the other hand, transcription inhibition significantly reduced the binding activity of homeoboxes in 8-cell embryos (Fig. 5). Altogether, maternal and embryonic factors appear to work together to establish the appropriate chromatin landscape for EGA.

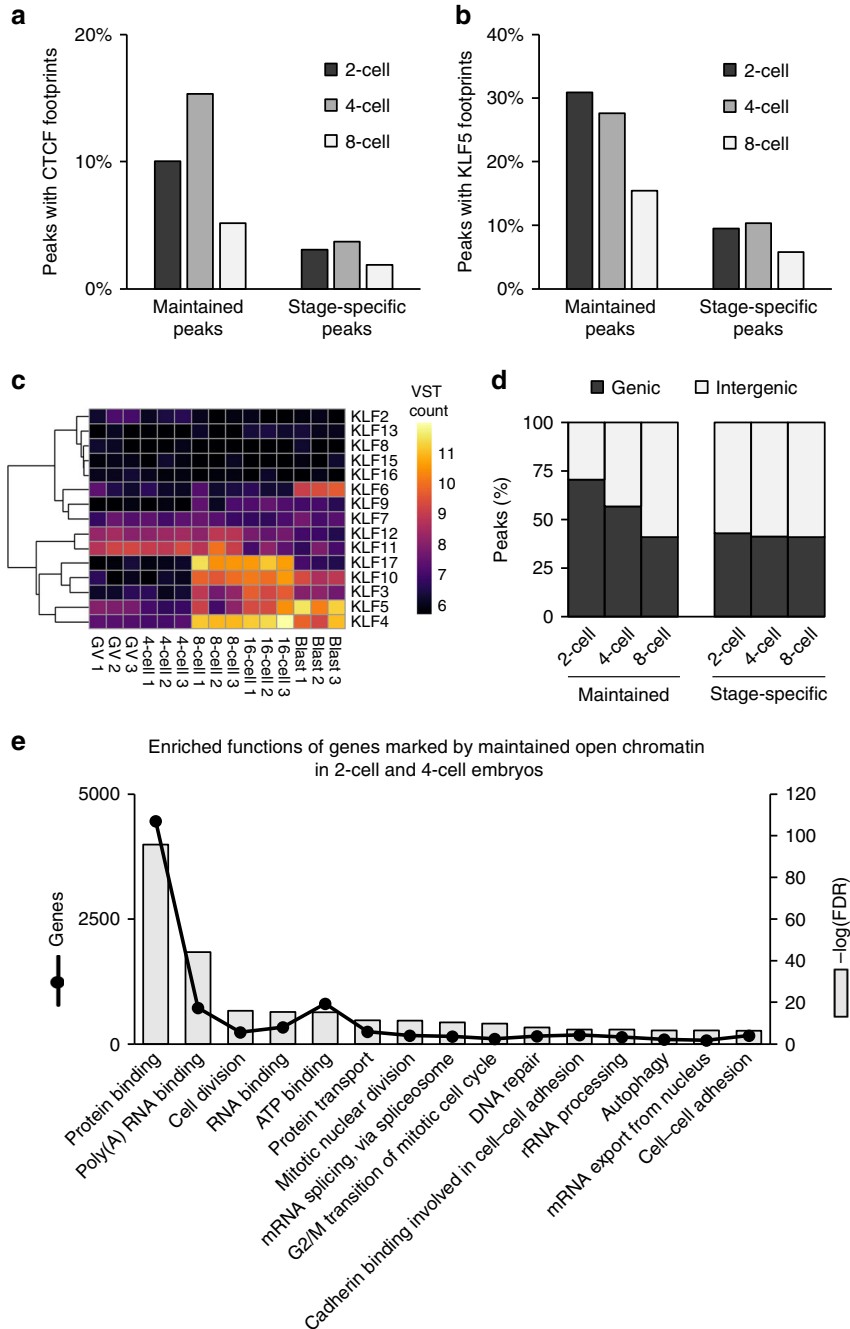

**Fig. 3 Binding motif enrichment in 2-, 4-, and 8-cell ATAC-seq peaks that were either stage-specific or subsequently maintained up to morula stage.**
**Proportion of peaks containing.** **a** CTCF or (**b**) KLF5 footprints. **c** VST expression profiles of *KLF* genes. RNA-seq data from Graf et al.[32]. **d** Proportion of maintained and stage-specific peaks that were genic or intergenic. **e** Gene ontology term enrichment of the 9,456 genes that were marked by maintained genic open chromatin that was first established in 2- or 4-cell embryos. Top 15 terms, based on FDR, are reported. Source data are provided as a Source Data file.

**Repetitive element enrichment in embryonic open chromatin.**
Notably, the MZT is not just characterized by a shift in the transcriptome, but also in the repeatome. Over 40% of the bovine genome is comprised of repeats derived from retrotransposons[59]. These repeats propagate through a "copy and paste" mechanism, and their expression is generally suppressed to avoid deleterious integrations[60]. However, retrotransposons are actively transcribed in preimplantation embryos, and although this phenomenon was previously considered opportunistic expression due to an unusually permissive chromatin state, the activity of some retrotransposons is actually crucial for development[61]. Although the

specific mechanisms behind this necessity are still being investigated, transposable elements have been implicated at several regulatory levels, as they can provide binding sites for TFs, behave as alternative promoters and enhancers[62] and contribute to 3-D chromatin architecture[63].

A complete catalog of repeat transcription throughout bovine preimplantation development was lacking. To address this gap in knowledge, available RNA-seq data[32] were assessed for expression of repetitive elements. Importantly, these libraries were not subjected to polyA selection. As has been observed in mouse and human[64,65], expression and accessibility of repetitive elements

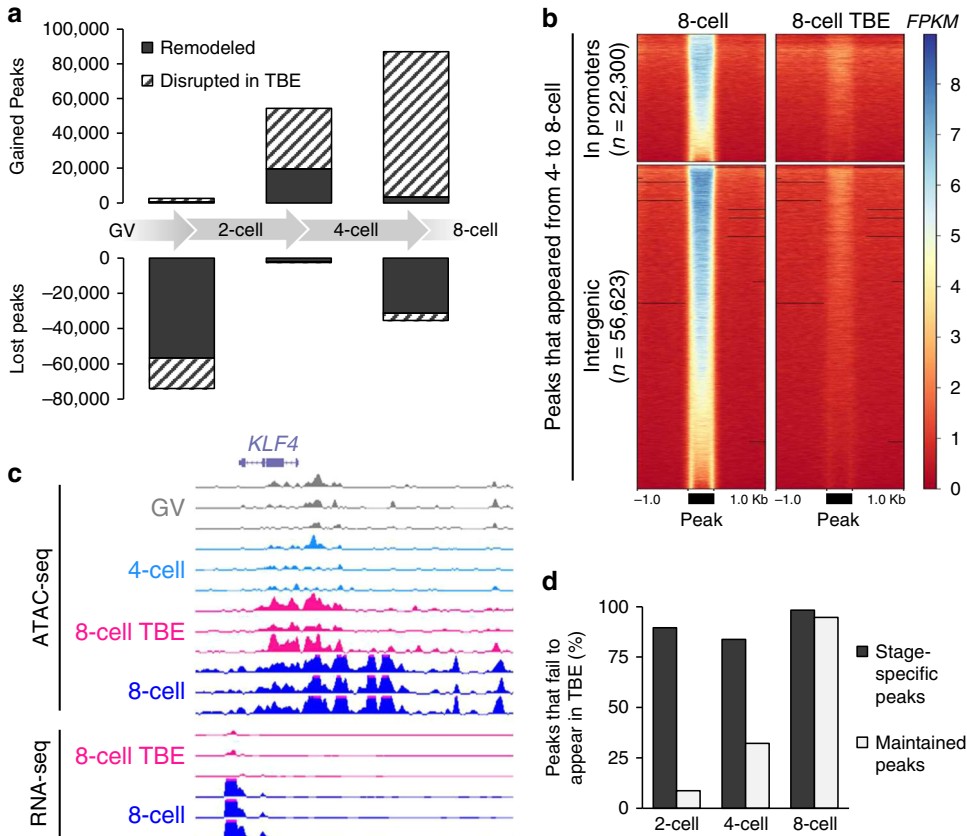

**Fig. 4 Effect of transcription inhibition on chromatin remodeling. a** Transcription inhibition effect on loci that should have opened or closed between consecutive stages. Peaks called from 20 million reads per stage. **b** Normalized ATAC-seq signal (FPKM) in control and transcription inhibited 8-cell embryos at peaks which were gained between the 4- and 8-cell stage. Peaks scaled to 500 bp. **c** Normalized ATAC-seq (FPKM) and RNA-seq signal (reads per kilobase million; RPKM) in 8-cell control and TBE at the *KLF4* locus. RNA-seq data from Bogliotti et al.[23]. **d** Proportion of stage-specific and maintained peaks that should appear at each stage, but which fail to appear in TBE. Source data are provided as a Source Data file.

throughout bovine preimplantation development was highly stage-specific and dynamic (Fig. 6a; Supplementary Fig. 12).

Non-long terminal repeat retrotransposons—long interspersed elements (LINEs) and short interspersed elements —are increasingly transcribed during human preimplantation development[19,64]. In cattle, LINEs also demonstrated increasing expression (Fig. 6a) and accessibility (Fig. 6b) starting at the 8-cell stage. Although the function of LINE elements in bovine preimplantation embryos has yet to be established, their activation is crucial in mouse for development to proceed past the 2-cell stage[66]. Notably, LTR activation occurs in human[64], mouse[65], and bovine preimplantation development[67]. In particular, mammalian LTRs and ERVLs were increasingly enriched in open chromatin starting at the 8-cell stage (Fig. 6b), although their expression dropped throughout development (Fig. 6a), indicating that other mechanisms likely regulate repeat expression, e.g., DNA methylation or histone modifications[68]. Only endogenous retroviral K (ERVK) elements demonstrated both increasing expression (Fig. 6a) and increased prevalence in distal open chromatin at the 8-cell stage (Fig. 6b). These patterns could indicate that ERVK elements are co-opted as regulatory elements in early embryos, as observed in humans[69], although the regulatory potential of these elements would need to be further explored and validated.

Mounting evidence suggests that specific LTR retrotransposons, especially intact elements, play pivotal roles in early development. During bovine preimplantation development, the most expressed retrotransposons were ERV1-1_BT and ERV1-2_BT (Fig. 7a). Upon further inspection, intact ERV1-1_BT

elements often co-occurred with MER41_BT repeats in a specific sequence, which demonstrated reproducible patterns of chromatin accessibility (Fig. 7b, Supplementary Fig. 13a). Furthermore, MER41_BT elements that were accessible in 8-cell embryos were enriched for the binding motifs of several pluripotency factors, including POU5F1, NFY, KLF4, OTX2, and TEAD (Supplementary Fig. 13b), suggesting that pluripotency factors may drive transposon expression.

In human and mouse pre-EGA embryos, DUX drives the expression of intact ERVL[21,24–26]. Specifically, in human embryos DUX4 binds MLT2A1 elements[21]. Considering that intergenic open chromatin was especially enriched for DUX binding sites in 8-cell bovine embryos, it seemed likely that these sites would also correspond to retrotransposons. Indeed, several MLT elements were enriched in 8-cell open chromatin harboring DUX motifs (Supplementary Table 6), suggesting that bovine DUX could be regulating the expression of LTRs with sequence similarity to MLT2A1 elements. Nearly all LTRs that were enriched in 8-cell open chromatin with DUX binding sites demonstrated dynamic expression profiles throughout development (Supplementary Fig. 14). In particular, MLT1A0 elements with DUX motifs experienced sharp increases in accessibility and expression in 8-cell embryos (Fig. 7c, d).

Whether LTRs actively contribute to bovine development simply as a consequence from opportunistic expression remains to be established. However, evidence in other species suggests that retrotransposons are often co-opted as promoters and enhancers[62]: a phenomenon that appears to extend to bovine embryos, as ATAC-seq and RNA-seq suggest that MLT elements are

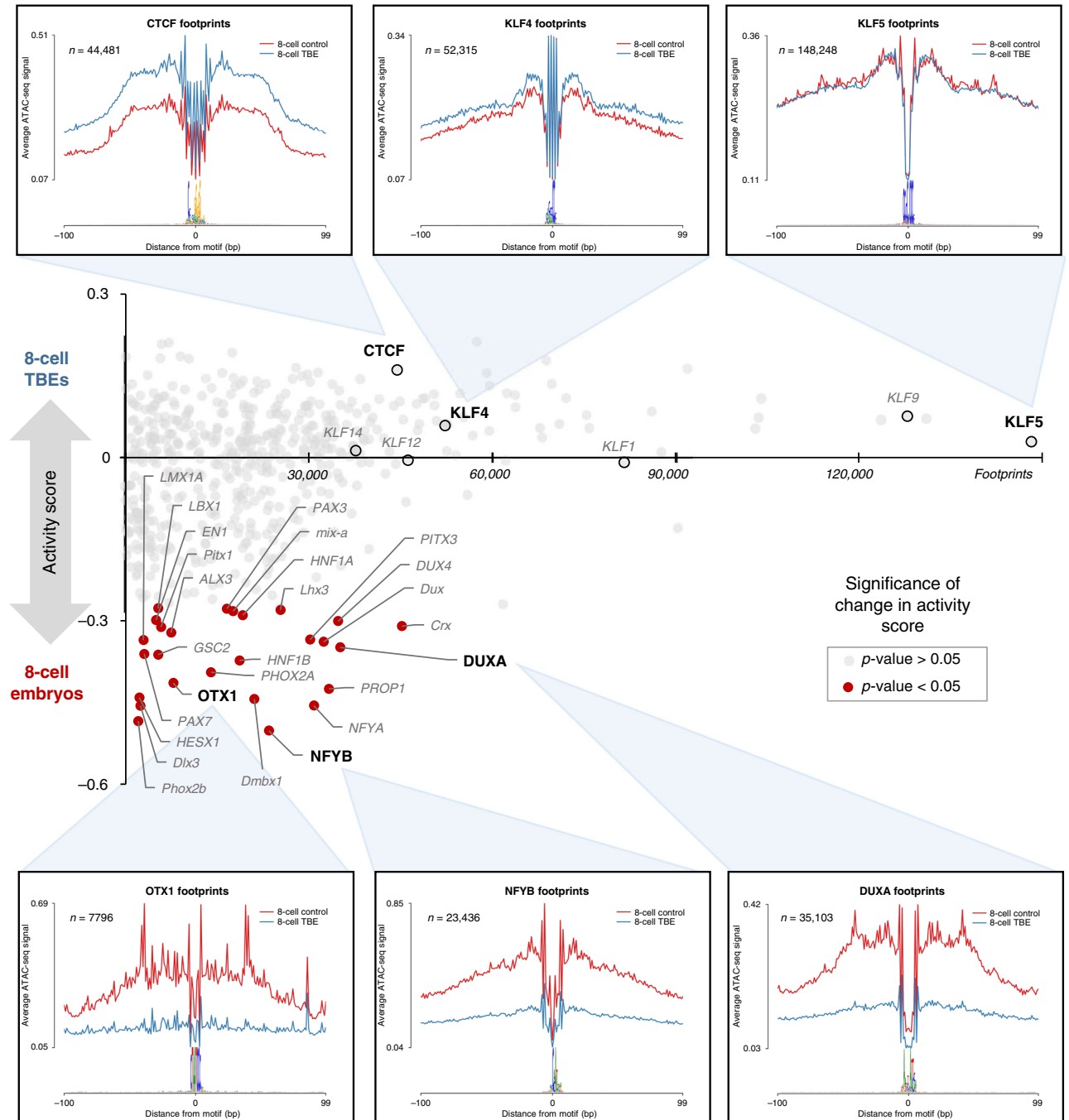

**Fig. 5 Differential transcription factor binding activity in 8-cell control and TBEs.** TFs marked in red demonstrated significantly reduced binding activity when embryonic transcription was inhibited ($p$ value < 0.05; two-tailed $t$ test). Average ATAC-seq signal around footprints is shown for select TFs, including three TFs with no significant change in binding activity (top) and three TFs with significantly reduced binding activity in TBEs (bottom). Source data are provided as a Source Data file.

co-opted as alternative promoters (Fig. 7e). An interesting balance appears to exist between repetitive elements and the embryo, wherein the repeatome leverages the embryo's existing regulatory network to drive transposition, while simultaneously providing novel regulatory elements and TF binding sites that the embryo co-opts to drive the expression of its own transcriptional programs.

**A model for mammalian genome activation.** Recent evidence suggests that regulation of the MZT may differ between mammals. First, the timing of genome activation is species-specific. Second, expression patterns of repetitive elements are not only stage-specific, but also species-specific; primate-specific and murine ERVL elements are strongly expressed during pre-implantation development[21,24–26]. Finally, the maternal programs in human and mice are divergent; human embryos conspicuously lack the murine maternal effect transcripts *POU5F1*, *HSF1*, and *DICER1*[70].

The relaxed chromatin structure in early preimplantation embryos provides a unique regulatory context for maternal factors, which are essential to support cleavage stage-embryos

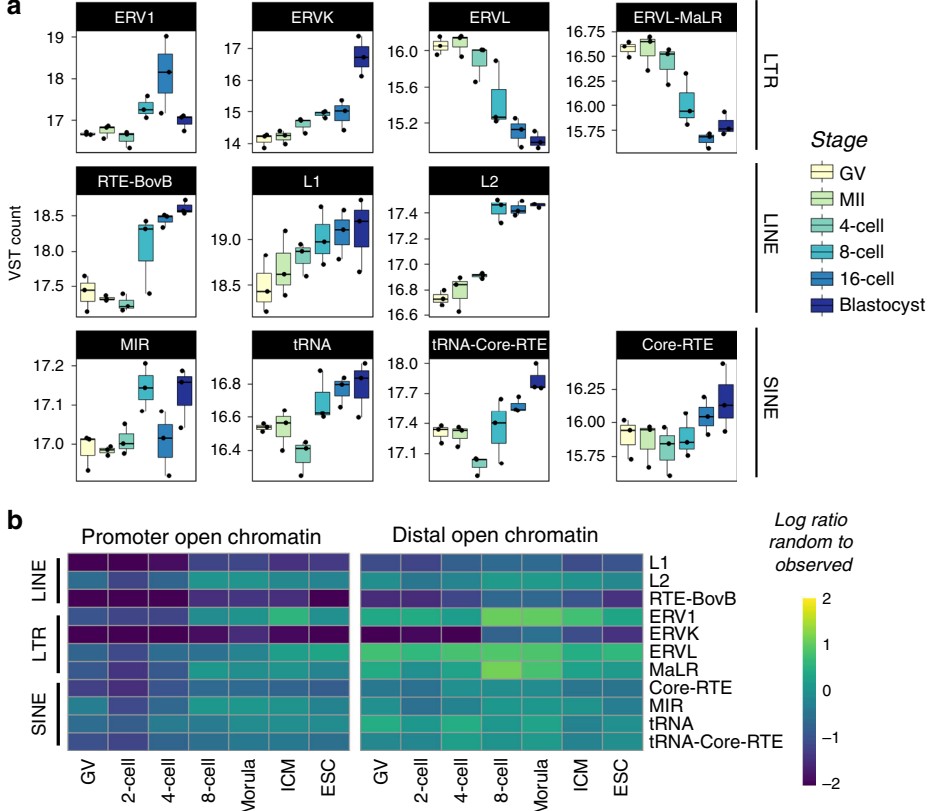

**Fig. 6 Expression and accessibility dynamics of repeat families during bovine preimplantation development.** Boxplots indicate the median and interquartile range (IQR), and whiskers span 1.5 times the IQR. **a** VST normalized expression profiles of select LTR, LINE, and SINE repeat families. RNA-seq data from Graf et al.[32]. **b** Enrichment of several transposable element (TE) families in ATAC-seq peaks, called from 30 million reads. Promoter peaks fell within the 2 kb region upstream of TSS. Intergenic peaks did not overlap the 2 kb regions upstream of TSS, exons, or introns ($n = 3$ biologically independent samples). Source data are provided as a Source Data file.

before EGA. Whereas the condensed chromatin structure in somatic cells generally restricts DNA-binding proteins to regions of open chromatin, the dispersed chromatin in 2-cell bovine embryos may allow maternal factors to opportunistically and pervasively bind their recognition motifs. Although the chromatin landscape changes markedly upon major EGA in bovine, human[18,19,21], and mouse[17,22] embryos, active regulators during this developmental stage appear to differ significantly between humans and mice[18], which suggests that regulation of mammalian EGA is species-specific. Genomic regulatory elements tend to evolve rapidly, making it difficult to identify homologous regions and compare accessibility in different species. On the other hand, sequences contained in open chromatin can indicate which regulatory factors are active in a given cell type. Therefore, the open chromatin landscapes of cattle, human, and mice embryos and ESC were compared for enrichment of known TF binding motifs. This analysis clearly clustered cattle and human together, especially 8-cell embryos and ESC, whereas the mouse remained distinct until later stages, when ICM and ESC regulatory circuitry began to converge with that of cattle and human ICM and ESC (Fig. 8a). To identify putative regulators of mammalian EGA, the enrichment of actively expressed TF motifs was compared across species and developmental stages. The activity of several regulatory factors differed between species, clearly separating humans and cattle from mice, and again suggesting divergence in regulatory and transcriptional programs (Supplementary Fig. 15). Compared to mouse 2-cell embryos, open chromatin in bovine and human 8-cell embryos demonstrated remarkably similar

patterns of sequence enrichment. In cattle and human embryos, SP1, OTX2, and NFY were particularly implicated during major EGA, whereas NR5A2, RARG, and ESRRB were solely implicated in mouse EGA (Fig. 8b).

Nevertheless, it remains unclear if these TFs are regulators of EGA, or simply products of it. For instance, although NR5A2 binding sites were enriched in 2-cell mouse embryos, NR5A2 is an early regulator of inner cell mass and trophectoderm programs and is not essential for genome activation[17]. Although several TFs appeared to only be important in either mouse, or in cattle and humans, KLFs were substantially enriched during EGA in all three species. Considering the well-established role of KLFs in somatic cell reprogramming and establishment of pluripotency[54], KLFs may play a conserved role in the MZT.

## Discussion
Sweeping changes to chromatin structure during bovine preimplantation development suggest that 2-cell embryos are characterized by globally decondensed chromatin, which is gradually compacted as development progresses, echoing similar observations in humans and mice. In particular, it is tempting to speculate that a conserved set of maternal factors establish basal promoter accessibility and the necessary chromatin architecture for enhancer-promoter interactions that will drive gene expression during major EGA (Fig. 9). However, the open chromatin landscape during major EGA clearly distinguished mice from cattle and humans, suggesting that whereas maternal regulation

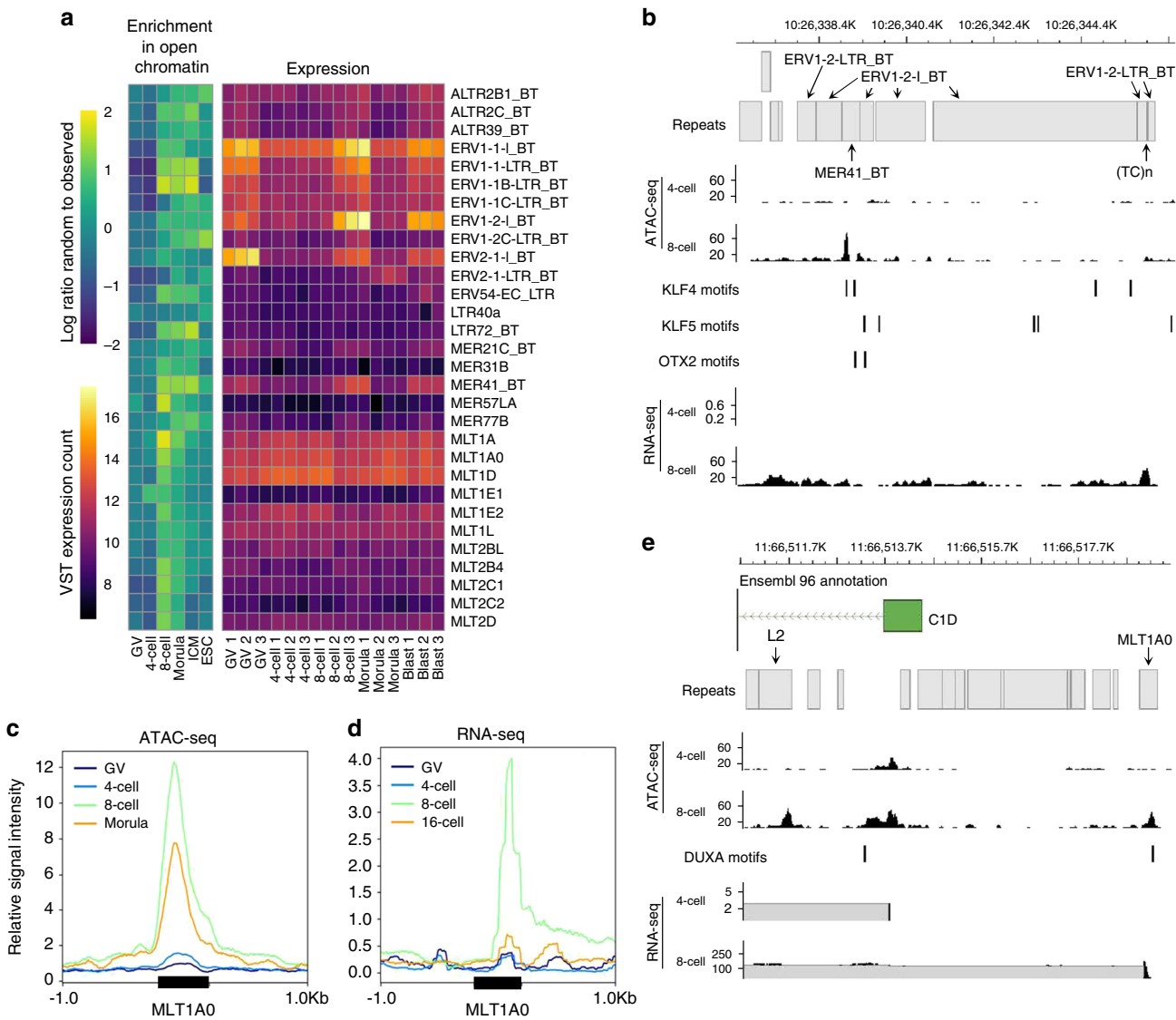

**Fig. 7 Activity of LTR elements during bovine preimplantation development. a** Open chromatin enrichment of LTR elements with increased accessibility in 4-cell, 8-cell, and/or morula-stage embryos, and VST expression of the same LTR elements; three replicates per stage. **b** ATAC-seq and RNA-seq read coverage at a highly expressed ERV1-2 repeat sequence. ATAC-seq tracks show coverage from 30 million reads per library; RNA-seq tracks show combined coverage from three replicates. TF motifs predicted from JASPAR binding motifs MA0039.3 (KLF4) and MA0712.1 (OTX2), respectively. **c** Average normalized ATAC-seq (FPKM) and (**d**) RNA-seq signal (RPKM) at MLT1A0 repeats overlapped by 8-cell intergenic open chromatin harboring DUXA motifs, predicted based on JASPAR motif MA0468.1. **e** Co-option of an accessible MLT1A0 element with a DUXA binding motif as an alternative promoter upstream of the *C1D* locus. RNA-seq data from Graf et al.[32]. Source data are provided as a Source Data file.

of EGA may be conserved across mammals, the transcriptional programs that are subsequently activated have diverged substantially. The resemblance of cattle and human EGA, both with respect to timing and associated regulators, strongly suggests that cattle are a more appropriate model system for human preimplantation development than mice. Nevertheless, the factors that appear to regulate the MZT in cattle, humans, and mice certainly warrant further investigation and validation, which will provide invaluable insight into the regulatory framework that governs successful preimplantation development.

## Methods

**Oocyte collection and maturation.** Ovaries were procured from a local abattoir and transported to the laboratory in a warm saline solution, complying with University of California Davis relevant ethical regulation for animal testing and research. Follicles measuring 2–10 mm were aspirated to obtain cumulus oocyte complexes (COCs). Only COCs with healthy layers of cumulus cells were selected for maturation. These were washed in collection medium 6:4 M199 (Sigma

M7653): SOF-Hepes, supplemented with 2% fetal bovine serum (FBS; Hyclone/ Thermo Scientific) and transferred to maturation medium modified M199 medium (Sigma M2154) supplemented with ALA-glutamine (0.1 mM), sodium pyruvate (0.2 mM), gentamicin (5 μg/ml), EGF (50 ng/ml), oFSH (50 ng/ml), bLH (3 μg/ml), cysteamine (0.1 mM), and 10% FBS.

**In vitro fertilization and embryo culture.** After COCs matured for 24 h, MII oocytes were washed in SOF-IVF medium (107.7 mM NaCl, 7.16 mM KCl, 1.19 mM $KH_2PO_4$, 0.49 mM, $MgCl_2$, 1.17 mM $CaCl_2$, 5.3 mM sodium lactate, 25.07 mM $NaHCO_3$, 0.20 mM sodium pyruvate, 0.5 mM fructose, 1X nonessential amino acids, 5 μg/ml gentamicin, 10 μg/ml heparin, 6 mg/ml fatty acid-free BSA) and transferred to drops of SOF-IVF medium under mineral oil. Frozen semen from a Holstein bull was thawed, and $10^6$ sperm/ml were added to drops with MII oocytes, which were incubated at 38.5 °C for 12–18 h. Zygotes were then removed from fertilization medium, and cumulus cells were removed by vortexing for 5 min in SOF-Hepes medium. Zygotes were then transferred to culture media (KSOMaa supplemented with 4 mg/mL BSA) under mineral oil, and incubated at 38.5 °C in 5% $CO_2$, 5% $O_2$, and 90% $N_2$. If embryos were to be transcriptionally inhibited, the culture medium was supplemented with α-amanitin (50 μg/ml) on day-one. On day-three, the culture medium was supplemented with 5% stem-cell qualified FBS

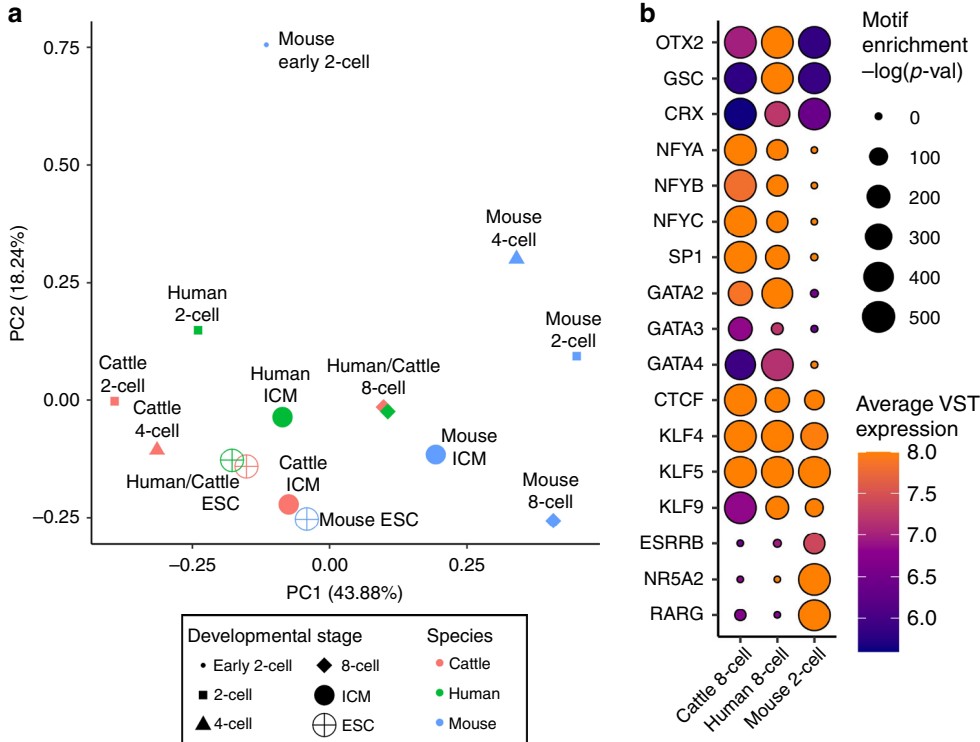

**Fig. 8 Comparison of enriched motifs in open chromatin during cattle, human, and mouse preimplantation development. a** PCA comparing the percent of ATAC-seq peaks that contained a given TF motif for each cell type and species. **b** Inference of key regulatory factors during EGA in cattle, human, and mouse, based on enrichment of TF binding motifs in open chromatin and expression of the corresponding TFs. For motif enrichment, binomial test *p* values reported. Bovine RNA-seq data from Graf et al.[32] human ATAC-seq and RNA-seq data from Wu et al.[18] mouse ATAC-seq and RNA-seq from Wu et al.[17]. Source data are provided as a Source Data file.

(Gemini Bio 100–525). Blastocyst development (minimum 30%) was evaluated at 7 days post-insemination (dpi).

**Culture of bovine ESC (bESC) and preparation for ATAC-seq.** CTFR-bESC lines derived from ICM through Wnt-pathway inhibition[37] were cultured in CTFR medium consisting in custom mTeSR1 (devoid of TGFb and FGF2) supplemented with 20 ng/mL human FGF2 (100–18B, Peprotech), 2.5 μM IWR-1 (I0161, Sigma Aldrich), and 100 U/mL penicillin–100 μg/mL streptomycin (15140–122, Gibco) on irradiated mouse embryonic fibroblast (MEF) (A34180, Gibco) at 37 °C and 5% CO₂. Cells were sub-cultured every 4–5 days at 1:10 split ratio using TrypLE Express (12604–013, Gibco). Culture medium was changed daily between passages. To increase cell survival, bESC culture medium was supplemented with 10 μM ROCK inhibitor Y-27632 (ALX-270-333-M005, Enzo) at seeding. Before cell lysis and library preparation CTFR-bESCs were trypsinized and separated from MEF feeders by plating the cell suspension in CTFR culture medium supplemented with 10 μM ROCK inhibitor Y-27632 for 1 h in a 60 mm cell culture dish (430196, Corning) at 37 °C and 5% CO₂. After incubation, cells that did not adhere to the culture dish were recovered for downstream processing. For each library, 50,000 cells were centrifuged at 500 rcf, 4 °C, for 5 min. Pellets were then washed once with 50 μl cold 1x PBS buffer and centrifuged at 500 rcf, 4 °C, for 5 min. Pellets were then resuspended in 50 μl cold ATAC-seq lysis buffer (10 mM Tris-HCl, pH 7.4, 10 mM NaCl, 3 mM MgCl₂ and 0.1% IGEPAL CA-630) and centrifuged at 500 rcf, 4 °C, for 10 min to isolate nuclear pellets.

**Preparation of oocytes and embryos for ATAC-seq.** Pools of oocytes and embryos were collected for ATAC-seq library preparation from three separate collections per developmental stage. Embryos intended for collection at the 2-cell, 4-cell, or 8-cell stages were divided into two groups, one of which was supplemented with α-amanitin, and cultured simultaneously. GV oocytes were collected for ATAC-seq prior to maturation. Preimplantation embryos were collected at the 2-cell (30–32 h post-insemination), 4-cell (2 dpi), 8-cell (3 dpi), and morula stages (5 dpi). For ICM collection, blastocysts were collected at 7 dpi and subjected to immunosurgery. Oocytes or embryos (a minimum of 500 cells) were treated with pronase (10 mg/ml) to completely remove the *zona pellucida* and washed with SOF-Hepes on a warming plate. Cells were then transferred to 1 ml cold SOF-Hepes, and centrifuged at 500 rcf, 4 °C, for 5 min. Morulae were vortexed for 3 min in cold ATAC-seq lysis buffer. Cell pellets were then resuspended in 1 ml cold ATAC-seq cell lysis buffer and centrifuged at 500 rcf, 4 °C, for 10 min, and

supernatant was aspirated to isolate nuclear pellets. To isolate ICMs by immunosurgery, blastocysts were treated with pronase (10 mg/ml) and washed in SOF-Hepes medium, then washed in 20% antibovine whole serum (B8270-2ML SIGMA, dilute in SOF-Hepes medium), and transferred to individual drops of 20% antibovine whole serum solution under mineral oil (20 ul per drop). Blastocysts were incubated for 1 h at 38.5 °C, washed in SOF-Hepes medium, then washed in 20% guinea pig complement (Innovative Grade US origin, IGP-COMPL-21249) and transferred to individual drops of 20% guinea pig complement solution under mineral oil (20 μl drop). Blastocysts were again incubated for 1 h at 38.5 °C. ICM were isolated by pipetting, treated with trypsin for 2–3 min, and pipetted several times before adding 10% fetal bovine serum to stop trypsinization. ICM cells were then washed in 1 ml cold SOF-Hepes medium and centrifuged at 500 rcf, 4 °C, for 5 min. Cell pellets were then resuspended in 1 ml cold ATAC-seq cell lysis buffer and centrifuged at 500 rcf, 4 °C, for 10 min, and supernatant was aspirated to isolate nuclear pellets.

**Transposition and ATAC-seq library construction.** Nuclear pellets were then resuspended in 50 μl transposition reaction mix (25 μl TD buffer (Nextera DNA Library Prep Kit, Illumina), 2.5 μl TDE1 enzyme (Nextera DNA Library Prep Kit, Illumina), 22.5 μl nuclease-free H₂O) and incubated for 60 min at 37 °C, shaking at 300 rpm. The transposase, which is loaded with Illumina sequencing adapters, cuts DNA where it is not sterically hindered and simultaneously ligates adapters, effectively producing a library in one incubation step. Transposed DNA was purified with the MinElute PCR purification kit (Qiagen, Hilden, Germany) and eluted in 10 μl buffer EB. Libraries were then PCR amplified: 50 μl reactions (25 μl SsoFast Evagreen supermix with low ROX (Bio-Rad, Hercules, CA), 0.6 μl 25 μM custom Nextera PCR primer 1, 0.6 μl 25 μM custom Nextera PCR primer 2 (containing the index used for demultiplexing; Supplementary Table 7), 13 μl nuclease-free H₂O, and 10 μl eluted DNA) were cycled at 72 °C for 5 min, 98 °C for 30 s, and then thermocycling at 98 °C for 10 s, 63 °C for 30 s and 72 °C for 1 min. Libraries from GV oocytes, 2-cell, and 4-cell embryos were thermocycled for 13 cycles, and 8-cell and morulae libraries were thermocycled for 11 cycles. PCR-amplified libraries were again purified with the MinElute PCR purification kit (Qiagen, Hilden, Germany) and eluted in 10 μl buffer EB. Libraries were then evaluated for DNA concentration and nucleosomal laddering patterns using the Bioanalyzer 2100 DNA High Sensitivity chip (Agilent, Palo Alto, CA). Expected nucleosomal laddering was evidenced by the presence of both small fragments, corresponding to hyper-accessible DNA that was frequently transposed, and larger fragments, corresponding to DNA that was wrapped around one or more

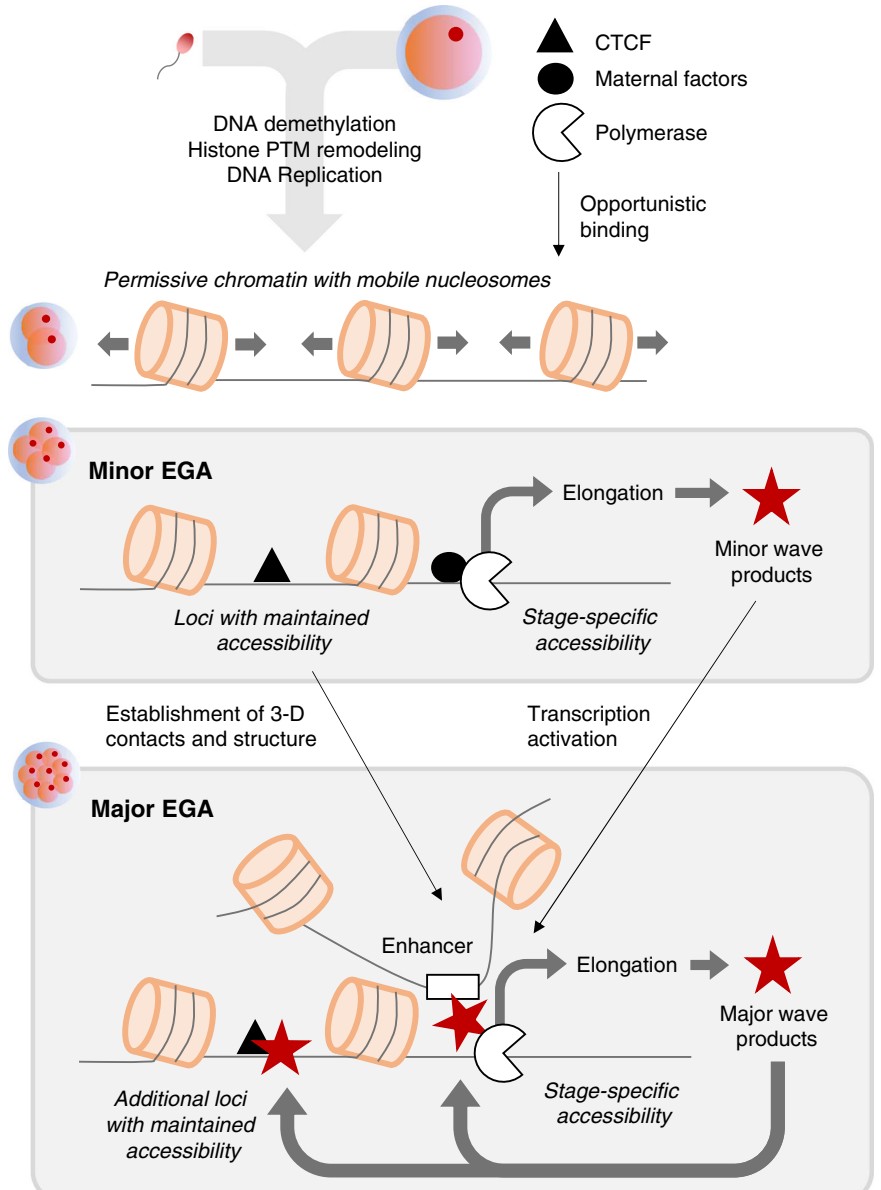

**Fig. 9 Potential mechanistic model depicting events leading to major EGA.** Chromatin structure is globally decondensed following fertilization, allowing opportunistic binding of maternal factors, which initiate a minor wave of transcription and begin to establish 3-D chromatin architecture. This sets the stage for major EGA, wherein maternal products, minor EGA products, and promoter-enhancer contacts mediate the first major wave of gene expression and continue to refine 3-D chromatin structure. PTM post-translational modifications.

nucleosomes. This study focused on mapping open chromatin; therefore, the sub-nucleosomal length fraction of each library (150–250 bp) was size selected using the PippinHT system (Sage Science, Beverly, MA) with 3% agarose cassettes. Size-selected libraries were run on a Bioanalyzer DNA High Sensitivity chip to confirm size-selection and determine DNA concentration. Final libraries were then pooled for sequencing on the Illumina NextSeq platform to generate 40 bp paired end reads.

**ATAC-seq read alignment and peak calling**. Raw sequencing reads were trimmed with Trim_Galore, a wrapper around Cutadapt (v0.4.0)[71], to remove residual Illumina adapter sequences and low quality ($q < 20$) ends, keeping unpaired reads and reads 10 bp or longer after trimming. Trimmed reads were then aligned to either the GRCm38 (mouse), GRCh38 (human), or ARS-UCD1.2 (cattle) assemblies using BWA aln (-q 15) and sampe[72]. PCR duplicates were removed with PicardTools, and mitochondrial and low-quality alignments ($q < 15$) were removed with SAMtools[73]. Alignments from biological replicates from each stage were merged and randomly subsampled to equivalent depth with SAMtools for detection of open chromatin. To determine which regions of the genome demonstrated significant enrichment of ATAC-seq signal, broad peaks were called with

MACS2[74], using a $q$ value cutoff of 0.05, and settings --nomodel --shift -100 --extsize 200.

**RNA-seq alignment and gene expression quantification**. Raw sequencing reads were trimmed with Trimmomatic (v0.33)[75]. Low-quality leading and trailing bases (3 bases) were clipped, and illumine adapter sequences were removed, allowing 2 seed mismatches, a palindrome clip threshold of 30, and a simple clip threshold of 10. Sliding window trimming was conducted with a window size of 4 bases, and a quality threshold of 15. Reads 36 bases or longer were retained after trimming. Trimmed reads were aligned to either the GRCm38 (mouse), GRCh38 (human), or ARS-UCD1.2 (cattle) assemblies with STAR (v2.7.2a) with options –outFilterScoreMinOverLread 0.85 and –seedSearchStartLmax 30. Low quality alignments ($q < 5$) were removed with SAMtools. Raw counts were calculated for genes in the Ensembl 96 annotations for each species with summarizeOverlaps, from the R package GenomicAlignments (v1.18.1)[76], using 'Union' mode and allowing fragments for paired end data. The DESeq2 R package (v1.22.2)[77] was used to identify differentially expressed genes, which were required to demonstrate a logFC >2 and an adjusted $p$ value < 0.05. Variance stabilizing transformation (VST) was used to normalize expression counts for visualization.

**Repeat expression quantification**. Trimmed reads were aligned to the ARS-UCD1.2 genome assembly with STAR (v2.7.0)[78] with options –outFilterMultimapNmax 100, --winAnchorMultimapNmax 100, and --two-passMode Basic. Raw expression values for individual repetitive elements were calculated for repeats in the RepeatMasker annotation for the ARS-UCD1.2 assembly (downloaded from the UCSC Genome Browser) using TEtoolkit[79] in 'multi' mode, which improves quantification of transposable elements transcripts by including ambiguously mapped reads. Raw expression values were normalized with the variance stabilizing transformation included in the DESeq2 R package (v1.22.2).

**Comparison of ATAC-seq and RNA-seq genome-wide signal**. For both ATAC-seq and RNA-seq data, alignments were converted to bigwig format using bam-Coverage from the DeepTools suite[80], which binned the genome into windows (50 bp default size) and calculated normalized signal in each window (fragments per kilobase million (FPKM) for paired-end data, and reads per kilobase million (RPKM) for single-end data). These bigwig files were used for track visualization with the UCSC genome browser. To compare bigwig files from replicate libraries on a genome-wide scale, the multiBigWigSummary function from Deeptools was used in bins mode (bin size of 500 bp). The output was then piped to the plotPCA function from DeepTools to generate the PCA plot (options –transpose, --log2, and –ntop 150000 for ATAC-seq libraries; options –transpose, --log2, and –ntop 100000 for RNA-seq libraries), and the plotCorrelation function to calculate the Pearson correlation coefficient between biological replicates. To assess average accessibility or expression at genomic intervals of interest, ATAC-seq normalized signal (FPKM in 50 bp bins) was visualized using the DeepTools computeMatrix (in scale-regions mode with option –skipZeros) and plotHeatmap functions. Regions of interest were scaled to 500 bp, and signal was also plotted for 1 kb up- and downstream.

**Comparison and classification of ATAC-seq peaks**. Peak sets from different stages were compared using the BEDtools intersect function[81], requiring a minimum of 1 bp overlap to consider a peak shared by both sets. Similarly, peaks were classified as genic if they overlapped either the 2 kb region upstream of a transcription start site (TSS), exons, or introns by 1 bp. Otherwise, peaks were considered intergenic.

**Motif enrichment**. Genomic regions were evaluated for binding motif enrichment using the findMotifsGenome.pl script from HOMER (v4.8)[82], using the exact sizes of the input genomic intervals (-size given). The most significant known or de novo motifs were reported, based on $p$ value. Known motifs that matched significantly enriched de novo motifs were reported if their match score exceeded 0.6. To compare motif instance in open chromatin at different stages and in different species, sets of ATAC-seq peaks were evaluated for the presence of HOMER's set of known TF motifs, measured as the percent of ATAC-seq peaks containing any given motif. To compare these data, PCA was performed with the prcomp function from the stats R package (v3.6.3), with options center = T and scale = F.

**Genome-wide motif location prediction**. Position-weight matrices were downloaded from the JASPAR database for TFs of interest[38]. Using the FIMO tool from the MEME suite (v5.0.4)[83], TF motif locations ($p < 1e-4$) were predicted genome-wide in the ARS-UCD1.2 genome assembly.

**Transcription factor footprinting analysis**. The HINT framework from the Regulatory Genomics Toolbox[84] was used to identify active TF binding sites from ATAC-seq data. When TFs are actively bound to DNA, transposase is sterically hindered, resulting in a small region, termed a "footprint," where read coverage abruptly drops. To maximize ability to detect footprints, reads from biological replicates were pooled together for each developmental stage and treatment group, and broad peaks were called from pooled reads with MACS2[74] using a $q$ value cutoff of 0.05, and settings --nomodel --shift -100 --extsize 200. For identifying footprints in an individual stage or treatment group, the rgt-hint command was used in footprinting mode to predict footprints within the corresponding ATAC-peaks, based on signal from pooled reads (commands –atac-seq and –paired-end). These footprints were matched to known motifs in JASPAR[38] with the rgt-motifanalysis command in matching mode. For comparing TF binding activity between developmental stages, or between control and TBEs, ATAC-seq peaks called in each of the two stages or treatment groups were first combined with BEDtools merge. Footprints were then identified with rgt-hint in this merged set of ATAC-seq peaks based on signal from pooled reads in each of the two stages or treatment groups. Footprints were matched to known motifs in JASPAR[38] with rgt-motifanalysis. Finally, the rgt-hint command in differential mode was used to compare the chromatin accessibility for each TF in each of the two stages or treatment groups. For each TF, the resulting binding activity score indicated whether the TF was more strongly bound in one group than the other, and the associated p-value indicated whether this change in binding activity between groups was significant (two-tailed $t$ test; $p < 0.05$).

**Repeat class enrichment in genomic intervals**. To determine if repetitive elements (either individual elements, families, or classes) were enriched in open chromatin, the number of ATAC-seq peaks overlapping a set of repetitive elements was compared to randomized intervals (ATAC-seq peak locations shuffled with BEDtools shuffle function) overlapping the same set of repetitive elements, yielding a log ratio of random to observed.

**Functional annotation enrichment analysis**. For gene sets of interest, the Ensembl gene IDs of human homologs were fetched using biomaRt[85]. These Ensembl IDs were submitted to DAVID (v6.8)[86,87] to identify enriched biological functions using medium stringency. Gene ontology terms (GOTERM_BP_DIRECT, GOTERM_CC_DIRECT, and GOTERM_MF_DIRECT) with a false discovery rate (FDR) <0.05 were reported.

**Reporting summary**. Further information on research design is available in the Nature Research Reporting Summary linked to this article.

## Data availability
The following published NGS data sets were used, and accessed through the NCBI GEO repository: for bovine oocytes and in vitro produced embryos, raw RNA-seq data were downloaded from accession number GSE52415[32]. For mouse and human preimplantation embryo ATAC-seq and RNA-seq data, raw sequencing files were downloaded from accession numbers GSE66390[17] and GSE101571[18], respectively. The ATAC-seq data produced in this study are available through the NCBI GEO repository under accession number GSE143658. All other relevant data supporting the key findings of this study are available within the article and its Supplementary Information files or from the corresponding author upon reasonable request. Source data are provided with this paper. A reporting summary for this Article is available as a Supplementary Information file.

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

## Acknowledgements
Funding for this project was provided by NIH grant HD070044 to P.J.R. and R.M.S. M. M.H. was supported by a NIFA National Needs Fellowship Grant (USDA-NIFA Competitive Grant Project no. 2014–38420–21796) and an Austin Eugene Lyons Fellowship. Semen used for in vitro embryo production was provided by Semex (Madison, WI). The authors thank the entire P.J.R. laboratory for contributions to embryo production and lively discussion.

## Author contributions
Conceived and designed the experiments: M.M.H., R.M.S., and P.J.R. Performed the experiments: X.M., C.Z., and M.M.H. Wrote the manuscript: M.M.H., R.M.S., and P.J.R.

## Competing interests
The authors declare no competing interests.
