## [Peer Review File · Nature Communications]

Reviewers' comments:

Reviewer #1 (Remarks to the Author):

Halstead et al. describe the chromatin accessibility landscapes of bovine oocytes, 2-, 4-, 8-cell and morula stage embryos using ATAC-seq analysis. In addition, they applied ATAC-seq to transcriptionally inhibited 2- to 8-cell embryos. The authors provide an informative dataset that will be of broad interest and mined by others. This is an important dataset to compare with those published in other organisms such as the mouse and human. There are several points below that need to be addressed to improve the impact of the paper and to further support the claims made in this interesting paper.

Their main findings are that:

1. Open chromatin is gained progressively throughout development with promoters gaining accessibility first and enhancers at later stages.
2. Embryonic transcription was not required for the appearance of cis open chromatin in 2- and 4-cell embryos but it is necessary to establish stage-specific and distal open chromatin, especially in 8-cell embryos. This suggests that both maternal and embryonic gene products participate in chromatin remodelling in a complementary fashion.
4. Similar EGA factors play roles in human and bovine embryos (e.g. OTX2, SP1), whereas murine EGA factors (RARG, NR5A2, ESRRB) are not enriched in the bovine embryo.
5. There are a set of conserved maternal regulators (e.g. CTCF, KLFs, SP1 and NFY) that play a role in chromatin remodelling in pre-EGA human, mouse and cow embryos.
6. Accessible chromatin in bovine embryos is enriched in repetitive elements suggesting that their activation may have a role in cow development.

The issues that need to be addressed are:

For Fig. 1b and Fig. S1 the authors bin the reference genome in 50bp windows and normalised within the window using RPKM.

- The ATAC-seq libraries produced 40bp paired-end reads, so the authors normalised with FPKMs not RPKMs.
- The RNA-seq data from Graf et al. produced 80bp single-end reads so in this case it is right to use RPKM units.
- In the FPKM/RPKM formula, did the authors use 0.05kb as "gene length"? They need to clarify this and make appropriate corrections if necessary.
- What was the rationale behind the choice of bin size? How do the plots change for larger windows?

Many will be interested in mining the datasets generated in this paper. However, with the absence of the blastocyst dataset this may limit the biological insights that can be gained within this window of development. While heterogeneity of cell types will be a concern at the blastocyst stage, nonetheless this is an important dataset to compare to the other data in this manuscript. Ideally this could be at the single cell level but this is not a requirement, even if the mural TE could be separated from the ICM/polar TE this this would also be informative, or the whole blastocysts if these options are not possible.

In general when describing enrichment of motifs it is unclear which genes could be putative direct targets and this information would be interesting to others and would help to make biological sense of the enrichment analysis. For example, the enrichment of GATA motifs in morula stage embryos is very interesting. The authors have a list of genes associated with this enrichment. Are any of these associated with trophoctoderm function?

One informative plot that would strengthen the authors' claim that "the open chromatin landscape during major EGA clearly distinguished mice from cattle and humans" would be a PCA integrating mouse, human and cow chromatin accessibility data (with a focus on peaks enriched for TF motifs). In this plot, pre-EGA stages should cluster together whereas post-EGA, human and cow samples should form a group located far from the mouse samples in PCA space.

The authors reason that the low enrichment of open chromatin peaks at the 2-cell stage may be a technical issue inherent to endonuclease-based assays. If this is the case, is the loss of accessibility discussed in Fig. 2a (from GV to 2-cell) a real biological signal or a technical artefact? What about gain of accessible sites from the 2-cell to the 4-cell stages?

For peak-calling, the authors merged biological replicates from each stage and randomly subsampled each merged sample to 30 or 20 million reads to minimise bias from sequencing depth. I understand the rationale behind this but I'm worried of the effects that random subsampling could have. I think that the authors should perform a test in which they, for example, subsample 100 times and show that roughly the same peaks are called using some measure of overlap (e.g. Jaccard similarity) for the $100(100-1)/2$ comparisons. If subsampling doesn't impact peak-calling, the distribution of overlaps should be close to 1.

For peak-calling, the authors used MACS2 on broad peak mode. Broad peaks are usually called for histone modification or features that cover entire gene bodies, whereas narrow peaks are preferred for TF binding. It would be interesting to see if the results associated with TF motif analysis change if the latter mode is used.

Related to the previous comment, peaks of open chromatin do not necessarily correspond to TF binding sites. Algorithms have been developed to detect TF footprints (pattern where an active TF binds to DNA and prevents Tn5 cleavage) and decrease the number of false positive TF binding sites. How does the motif analysis change if it is performed after footprinting?

In the "RNA-seq alignment and gene expression quantification" and "Repeat expression quantification" sections of the Methods, the authors say that gene counts were MLE-normalised using DESeq2. Maximum likelihood estimation (MLE) is not a normalisation method, it is a way to estimate gene-wise dispersion prior to the statistical inference of differential expression (see PMID:25516281). Then, the authors say that they used variance stabilising transformation (VST) for some analyses, which is a normalisation method. I suggest the authors revise these sections carefully, indicating the situations in which MLE (most certainly differential gene expression) and VST (most likely visualisation) were used.

Fig. 1e: What about the morula stage?

Figure 2a: What is the size of the depicted window with the peak in the middle? The authors must indicate this in the x-axis of the plot.

Was ATAC-seq data performed on a pooled collection of embryos or cells? I think this needs to be clarified in the results and in the methods section. If pooled or from a whole embryo the caveats need to be discussed/acknowledged clearly in terms of cellular heterogeneity particularly at the later stages of development in the discussion section.

In the discussion about the enrichment of the DUX motif during EGA the authors need to acknowledge a recent paper from Yi Zhang which is a robust experiment showing that Dux is dispensable for mouse development and EGA (PMID: 31133747). This questions the essential requirement of Dux in EGA and suggests possible redundancy or that other factors may be involved separate to DUXA.

With alpha-amanitin treatment is it clear when there is embryonic arrest? Is this between the 4-8 cells stage or slightly later?

In the paragraph starting on line 219 the authors discuss the enrichment of CTCF motifs, 3D chromatin architecture and promoter enhancer switching. Do the authors see evidence of promoter enhancer switching in their ATAC-seq datasets for CTCF target genes? Or is the resolution not good enough to distinguish this?

The abstract needs to be improved to make it more clear what are the novel findings and the wording needs to be tightened.

On line 67 is it clear whether “accessible sites are gradually established as development progresses”? This question is also related to the paragraph starting on line 119 where a highly permissive chromatin structure is assumed especially given the absence of chromocenters and this is consistent with the gradual decrease in accessibility based on methylation of CpG sites I think this statement as written contradicts that “accessible sites are gradually established”. It is also confusing why the authors conclude that “global chromatin relaxation: is a shared characteristic from zygote stage onward -based on what is written isn’t this the opposite – i.e. chromatin accessibility becomes more restricted as development progresses? The on line 166 the first gains in accessibility occur in 4-cell stage embryos. Altogether as written this is very confusing.

On line 78 PTM needs to be defined.

Line 252 KLF4 is misspelled.

Reviewer #2 (Remarks to the Author):

This is an interesting paper comparing the bovine early embryos to human and mouse as a comparative biology analysis and using ATAC-seq to investigate open regions as a proxy to transcription activities. The question of which control which between chromatin structure and transcription activity is not fully resolved but discussed in length. Although a large part of the discussion is speculative, the postulates bring interesting hypothesis to be tested by knock-down experiments. The conclusion does not include any comments in relation to failure of EGA in many human and bovine embryos. One must realize that the pool of embryos used here were not left in culture to see if they would achieve the later stage creating a discontinuity in the observations. Failure to EGA is associated with poor oocyte quality and the value of the bovine model (high and low quality available) to resolve the important factors in the induction of EGA could be quite important if analyzed properly.

Line 33: combine following the first round of replication

Line 74 Confusing suggestion: 2-cell stage¹ in mice, during the 4- to-8-cell stage in humans²⁹ and pigs³⁰, and between the 8- and 16-cell stages in sheep³¹ and cattle³²

Line 116 Please clarify the filter of minimum number of reads to be considered in the differential analysis : To minimize bias from sequencing depth, peaks were called from either 30 million monoclonal uniquely mapped reads when comparing different developmental stages, or 20 million reads when comparing TBE with control embryos.

Line 132: CpG sites

Line 134 Please specify on which regions the methylation is associated with genome-wide accessibility in human embryos actually decreases from the zygote stage onward (ref 17)

Line 141. This is quite speculative suggesting that chromatin remodeling serves two functions: progressive establishment of a “totipotent status, as two cells are already recognized as totipotent. And then what about the DNA methylation removal as the indicator of totipotency, what comes first ?

line 183 There are multiple forms of NFkB and some are indeed peaking at the 1 cell stage in bovine RNAseq data and such could be added. (of course, a knock down would be nice to support the speculation)

Line 252 KLF4 ?

Figure 5A. the X axe is not defined (and not the same as 5B, with different number of stages?)

Line 332 Any idea if the POU5F1 KO results in less retro-elements to be produced (cow or mice)?

Line 373: I agree that maternal factors may have an initiator role in EGA and oocytes associated with

specific conditions in bovine results in a significant increase of embryos achieving development competence and hence passing the 8-cell stage. Does the stored RNAs (like KLF4) in this high competent group (like OPU after FSH or in vivo early embryos) could indicate which factors would act on the early openings of the 2 cell chromatin ?

Reviewer #3 (Remarks to the Author):

Halstead et.al. used bovine as a model to investigate the chromatin remodeling during early embryogenesis. They used ATAC method to map chromatin accessibility in genome wide. They proposed that bovine could be a good model to mimic the chromatin remodeling for human early embryogenesis. The major issue is that the results shown in the paper are just observation. Almost no causality mechanism can be drawn from their results, while the authors overstated their results throughout their paper.

1. Line 155. "The first major transition in chromatin structure mostly involved loss of hyperaccessible sites in 2-cell...". The authors did not examine the zygotes, they cannot know whether the chromatin states in zygotes are the same as that in 2-cell stage. If they want to make this conclusion, they should map the chromatin state of zygotes and demonstrate that the zygotic pattern is similar to 2-cell pattern.

2. Line 177. The author conclude that NFkB could play important role in activate major EGA. The conclusion is based on the higher binding frequency in 4- to 8-cell stage than other stage. This is just a correlation. If the author want to make the conclusion, they should knock-down NFkB and check its affect on EGA.

Line 192. The author conclude that DUXA play a conserved role during preimplantation development for placental mammals. The conclusion is based on the increased accessiblity during bovine development. This is also just a correlation. No causality result has been presented.

These are just two examples of the overstatements shown in this paper. Too many similar overstatements can be found throughout the paper.

3. Blastocyst is an important stage before implantation. The author should check that stage, and compare it with morula stage.

4. The author conclude that maternal and embryonic transcription drive chromatin reorganization. This is the only case showing causality.

5. Line 313. The author suggest that LINE2 elements could act as enhancers or promoter since these elements express and have accessiblity in 8-cell stage. I could not see any relationship between the data and their conclusion. Many similar statements can be found in the transposon section.

Minor comment:

Line 137. "Slate" should be state.

Poin-by-point response to reviewer's comments:

Reviewer #1: For Fig. 1b and Fig. S1 the authors bin the reference genome in 50bp windows and normalised within the window using RPKM.

- The ATAC-seq libraries produced 40bp paired-end reads, so the authors normalised with FPKMs not RPKMs.
- The RNA-seq data from Graf et al. produced 80bp single-end reads so in this case it is right to use RPKM units.
- In the FPKM/RPKM formula, did the authors use 0.05kb as "gene length"? They need to clarify this and make appropriate corrections if necessary.
- What was the rationale behind the choice of bin size? How do the plots change for larger windows?

Response: The reviewer is correct about the distinction between paired and single-end data. RPKM has been changed to read FPKM in Figures 1, 2, 4, and 6. The calculation of FPKM/RPKM was carried out by the bamCoverage function from the DeepTools suite, which was used to calculate the normalized signal in 50 bp windows. This is the default window size for this function, which was specifically designed for analysis of high-throughput sequencing data, so the default window size was not changed. This normalized signal (in 50bp windows) was used for visualization with the UCSC genome browser. For comparison of replicate libraries (PCA and correlation analyses), we adjusted the comparison bin size to 500 bp, and this significantly improved sample clustering in PCA plots, as well as correlations, so we thank the reviewer for this suggestion. These changes are indicated in methods section.

Reviewer #1: Many will be interested in mining the datasets generated in this paper. However, with the absence of the blastocyst dataset this may limit the biological insights that can be gained within this window of development. While heterogeneity of cell types will be a concern at the blastocyst stage, nonetheless this is an important dataset to compare to the other data in this manuscript. Ideally this could be at the single cell level but this is not a requirement, even if the mural TE could be separated from the ICM/polar TE this this would also be informative, or the whole blastocysts if these options are not possible.

Response: To address this concern, chromatin accessibility was profiled in inner cell mass (3 replicates) as well as embryonic stem cells (3 replicates).

Reviewer #1: In general when describing enrichment of motifs it is unclear which genes could be putative direct targets and this information would be interesting to others and would help to make biological sense of the enrichment analysis. For example, the enrichment of GATA motifs in morula stage embryos is very interesting. The authors have a list of genes associated with this enrichment. Are any of these associated with trophectoderm function?

Response: To identify putative gene targets, we looked for promoters with ATAC-seq peaks that contained transcription factor footprints. This revealed that at the morula stage, GATA footprints were found in the promoters of 1,031 genes that were functionally enriched for negative regulation of cell

differentiation. A summary of putative gene targets, based on TF footprints, can be found in Table S4.

Reviewer #1: One informative plot that would strengthen the authors' claim that "the open chromatin landscape during major EGA clearly distinguished mice from cattle and humans" would be a PCA integrating mouse, human and cow chromatin accessibility data (with a focus on peaks enriched for TF motifs). In this plot, pre-EGA stages should cluster together whereas post-EGA, human and cow samples should form a group located far from the mouse samples in PCA space.

Response: This was an interesting and challenging suggestion. Because regulatory elements tend to evolve rapidly, it is often difficult to identify homologous regions in other species, and therefore impossible to determine if chromatin accessibility is "conserved." This is especially true for enhancers, which are largely species-specific (Yue et al, 2014; Villar et al, 2015). Interestingly, several studies have demonstrated that enhancer function can be conserved, even when overall sequence is not (Hare et al, 2008; Fisher et al, 2006; Ludwig et al, 2000). In fact, it has been suggested that selective pressure may only operate on the functional components of regulatory elements: transcription factor binding sites (TFBS). Sequence conservation can be inferred from sequences as short as 36 bp (Villar et al, 2014), but conserved TFBS (6-12 bp (Tuğrul et al, 2015)) generally are not enough to identify homologous regulatory regions. Indeed, when we attempted to map ATAC-seq peaks detected in cattle to human and mouse, only 21% could be mapped to all three species.

Considering that this analysis would only account for a fraction of accessible sites, we instead decided to focus on TF motif presence in open chromatin. The motif enrichment analysis conducted by HOMER provided insight into the presence of known TF motifs in ATAC-seq peaks for each developmental stage and species. Therefore, PCA was used to integrate these data, comparing the percent of ATAC-seq peaks that contained a given TF motif for each cell type and species. This PCA clearly clustered cattle and human together (especially 8-cell embryos and ESC), whereas mouse remained distinct until later stages, when ICM and ESC regulatory circuitry began to converge with that of cattle and human ICM and ESC. See line 432 and Figure 8A.

Reviewer #1: The authors reason that the low enrichment of open chromatin peaks at the 2-cell stage may be a technical issue inherent to endonuclease-based assays. If this is the case, is the loss of accessibility discussed in Fig. 2a (from GV to 2-cell) a real biological signal or a technical artefact? What about gain of accessible sites from the 2-cell to the 4-cell stages?

Response: Rather than considering peaks that were lost from the GV to 2-cell stage, we instead focused our analysis on ATAC-seq peaks that were unique to GV oocytes. Furthermore, to avoid confusion we changed the wording throughout the text from gain or loss of accessibility, to gain or loss of ATAC-seq peaks.

Reviewer #1: For peak-calling, the authors merged biological replicates from each stage and randomly subsampled each merged sample to 30 or 20 million reads to minimise bias from sequencing depth. I understand the rationale behind this but I'm worried of the effects that random subsampling could have. I think that the authors should perform a test in which they, for example, subsample 100 times and show that roughly the same peaks are called using some measure of overlap (e.g. Jaccard similarity)

for the $100(100-1)/2$ comparisons. If subsampling doesn't impact peak-calling, the distribution of overlaps should be close to 1.

Response: To address this concern, we first established a baseline for peak set similarity by comparing biological replicates in our data, as well as human and mouse data previously published in Nature (Wu et al, 2016, 2018). Using the Jaccard statistic as a measure for similarity of peak sets, the consistency of our replicates was comparable or superior to the human and mouse data (Figure A). For the stages where random subsampling was implemented, informative reads for each stage and condition were merged and randomly subsampled 100 times to a depth of either 20 or 30 million reads, from which broad peaks were called. Jaccard statistics were then calculated for all pairwise comparisons. Random subsampling appears to yield more consistent sets of peaks than those derived from biological replicates. We conclude that subsampling does not introduce additional variation to the data. See line 141.

Reviewer #1: For peak-calling, the authors used MACS2 on broad peak mode. Broad peaks are usually called for histone modification or features that cover entire gene bodies, whereas narrow peaks are preferred for TF binding. It would be interesting to see if the results associated with TF motif analysis change if the latter mode is used.

Response: Depending on the properties of underlying regulatory elements, regions of ATAC-seq enrichment can be variably narrow or broad. As such, there is inconsistency in the literature regarding how ATAC-seq peaks should be called (i.e. whether MACS2 is the best peak caller, whether narrow or broad peaks are more appropriate, etc). For example, the developers of ATACseqQC call broad peaks with MACS2 (Ou et al, 2018), whereas the human preimplantation embryo ATAC-seq data that was re-analyzed in this manuscript was subjected to narrow peak calling with MACS2 (Wu et al, 2018). Regardless, MACS2 identifies broad peaks by grouping nearby enriched regions (narrow peaks) into broad regions. Calling narrow peaks instead of broad peaks would probably alter the TF motif analysis, but we certainly do not exclude any accessible motifs by calling broad peaks.

Reviewer #1: Related to the previous comment, peaks of open chromatin do not necessarily correspond to TF binding sites. Algorithms have been developed to detect TF footprints (pattern where an active TF binds to DNA and prevents Tn5 cleavage) and decrease the number of false positive TF binding sites. How does the motif analysis change if it is performed after footprinting?

Response: Following the reviewer's suggestion footprinting analysis was conducted for each stage of development. This analysis provided substantial insight into the active regulatory circuitry during preimplantation development, especially when the binding activity of TFs was compared between developmental stages (Figure 4), as well as between control and TBEs (Figure 5, Figure S10).

Reviewer #1: In the "RNA-seq alignment and gene expression quantification" and "Repeat expression quantification" sections of the Methods, the authors say that gene counts were MLE-normalised using DESeq2. Maximum likelihood estimation (MLE) is not a normalisation method, it is a way to estimate gene-wise dispersion prior to the statistical inference of differential expression (see PMID:25516281). Then, the authors say that they used variance stabilising transformation (VST) for some analyses, which is a normalisation method. I suggest the authors revise these sections carefully, indicating the situations in which MLE (most certainly differential gene expression) and VST (most likely visualisation) were used.

Response: All figures depicting expression data now show VST counts. Maximum likelihood estimation was only used to infer differential expression. The methods section for RNA-seq alignment and gene expression quantification has been clarified.

Reviewer #1: Fig. 1e: What about the morula stage?

Response: This figure focused on peaks that were either stage-specific, or which were established and then maintained up until the morula stage. Including morula as a stage in this graphic would not be informative, because every peak that was present at the morula stage would be maintained up until the morula stage.

Reviewer #1: Figure 2a: What is the size of the depicted window with the peak in the middle? The authors must indicate this in the x-axis of the plot.

Response: The x-axis of the plot has been updated to reflect the scale of the depicted window, and the figure caption also indicates this scale. Peaks were scaled to 500bp, and the 1kb region on either side is also depicted.

Reviewer #1: Was ATAC-seq data performed on a pooled collection of embryos or cells? I think this needs to be clarified in the results and in the methods section. If pooled or from a whole embryo the caveats need to be discussed/acknowledged clearly in terms of cellular heterogeneity particularly at the later stages of development in the discussion section.

Response: Yes, ATAC-seq was performed on pooled collections of embryos or cells. This has been clarified in both the results/discussion and methods sections. See line 121 and line 509.

Reviewer #1: In the discussion about the enrichment of the DUX motif during EGA the authors need to acknowledge a recent paper from Yi Zhang which is a robust experiment showing that Dux is dispensable for mouse development and EGA (PMID: 31133747). This questions the essential requirement of Dux in EGA and suggests possible redundancy or that other factors may be involved separate to DUXA.

Response: This paper is cited alongside similar publications on line 224, and mentioned later in line 422.

Reviewer #1: With alpha-amanitin treatment is it clear when there is embryonic arrest? Is this between the 4-8 cells stage or slightly later?

Response: In the presence of alpha-amantin, bovine embryos do not develop past the 8- to 16-cell stage (Memili et al, 1998). This detail has been added to line 303 of the text.

Reviewer #1: In the paragraph starting on line 219 the authors discuss the enrichment of CTCF motifs, 3D chromatin architecture and promoter enhancer switching. Do the authors see evidence of promoter enhancer switching in their ATAC-seq datasets for CTCF target genes? Or is the resolution not good enough to distinguish this?

Response: To our understanding, promoter-enhancer switching refers to a phenomenon wherein a regulatory element may act as a promoter or an enhancer, depending on phenotype. This concept is not discussed in this manuscript. If the reviewer is referring to the statement: "The global reorganization of 3D chromatin architecture in mouse has also been observed in bovine embryos, wherein gene-rich regions switch from a random distribution to a chromosome-specific distribution during major EGA," we have reworded this sentence to make our meaning more clear: "the non-radial establishment of chromosome territories, when gene-poor chromosomes tend to localize to the nuclear periphery more often than gene-rich regions, is first established at the 8-cell stage" (line 269).

Reviewer #1: The abstract needs to be improved to make it more clear what are the novel findings and the wording needs to be tightened.

Response: The abstract has been rewritten to better summarize the novel findings.

Reviewer #1: On line 67 is it clear whether "accessible sites are gradually established as development progresses"? This question is also related to the paragraph starting on line 119 where a highly permissive chromatin structure is assumed especially given the absence of chromocenters and this is consistent with the gradual decrease in accessibility based on methylation of CpG sites I think this statement as written contradicts that "accessible sites are gradually established". It is also confusing why the authors conclude that "global chromatin relaxation: is a shared characteristic from zygote stage onward -based on what is written isn't this the opposite – i.e. chromatin accessibility becomes more restricted as development progresses? The on line 166 the first gains in accessibility occur in 4-cell stage embryos. Altogether as written this is very confusing.

Response: To address this concern, we instead refer to the gain or loss of ATAC-seq peaks, rather than gain or loss of accessibility throughout the text. Additionally, the manuscript states that “Global chromatin relaxation appears to be a shared characteristic of human, bovine, and murine pre-EGA embryos” (line 173), and not that chromatin relaxation is a characteristic that persists from the zygote stage onward.

Reviewer #1: On line 78 PTM needs to be defined.

Response: This has been changed to read “post-translational modifications.”

Reviewer #1: Line 252 KLF4 is misspelled.

Response: This has been fixed.

Reviewer #2: Line 33: combine following the first round of replication

Response: The abstract has been reworded.

Reviewer #2: Line 74 Confusing suggestion: 2-cell stage¹ in mice, during the 4- to-8-cell stage in humans²⁹ and pigs³⁰, and between the 8- and 16-cell stages in sheep³¹ and cattle³²

Response: We modified the sentence to be less confusing by only including mouse, human, and cattle (see line 97), and now refer to evidence from more recent RNA-seq studies.

Reviewer #2: Line 116 Please clarify the filter of minimum number of reads to be considered in the differential analysis : To minimize bias from sequencing depth, peaks were called from either 30 million monoclonal uniquely mapped reads when comparing different developmental stages, or 20 million reads when comparing TBE with control embryos.

Response: Reads were randomly subsampled prior to peak calling, either to 20 million reads or 30 million reads, depending on the comparison. This has been clarified (line 138) : “Pooled reads from replicates were randomly subsampled to standard depths before calling peaks in order to minimize bias from sequencing depth. Peaks were called from either 30 million monoclonal uniquely mapped reads when comparing control embryos at different developmental stages, or 20 million reads when comparing TBE with control embryos.”

Reviewer #2: Line 132: CpG sites

Response: This was not a typo. The single-cell COOL-seq technique employs a GpC methyltransferase to mark accessible GpC sites. The method is detailed in a paper by Fan Guo et al (2017). PMID: 28621329. A diagram of the technique (image from paper previously referenced) is copied below. The description has been clarified at line 155: “a recent study that profiled chromatin accessibility with a technique that was

not based on endonuclease activity, but rather on GpC methyltransferase accessibility to DNA, found that ...”

Reviewer #2: Line 134 Please specify on which regions the methylation is associated with genome-wide accessibility in human embryos actually decreases from the zygote stage onward (ref 17)

Response: The referenced study used a GpC methylase, after cell lysis, to indirectly measure chromatin accessibility by methylating accessible GpC sites. The technique measures DNA methylation simultaneously, but overall found an inverse relationship between chromatin accessibility and DNA methylation. Therefore, the paper did not provide a description of regions where DNA methylation was associated with chromatin accessibility. Some figures from reference 17 (Fan Guo et al, 2017) are included below.

b**d**
Reviewer #2: Line 141. This is quite speculative suggesting that chromatin remodeling serves two functions: progressive establishment of a “totipotent status, as two cells are already recognized as totipotent. And then what about the DNA methylation removal as the indicator of totipotency, what comes first ?

Response: The sentence has been rephrased: “chromatin remodeling serves two functions: progressive establishment of a chromatin landscape that will support further development, and transient stage-specific regulation.” (line 167). DNA methylation removal is mentioned at line 75 of the text: “Following fertilization, the zygotic genome is globally demethylated (Iurlaro et al, 2017; White et al, 2016), and this loss of DNA methylation coincides with global decreases in repressive histone modifications (Eckersley-Maslin et al, 2018; Xu & Xie, 2018; Ross et al, 2008).” Clearly, 2-cell embryos are considered totipotent, and considering the wide-spread removal of DNA methylation by this stage of development, this could certainly be considered an indicator of totipotency.

Reviewer #2: line 183 There are multiple forms of NFkB and some are indeed peaking at the 1 cell stage in bovine RNAseq data and such could be added. (of course, a knock down would be nice to support the speculation)

Response: It would definitely be of interest to knock down NFkB factors and study the effect on embryonic development. We certainly plan to pursue this line of study in the future.

Reviewer #2: Line 252 KLF4 ?

Response: This has been corrected.

Reviewer #2: Figure 5A. the X axe is not defined (and not the same as 5B, with different number of stages?)

Response: The legend for this figure has been displayed more prominently. The different colors of the boxplots are indicative of different stages of development, to avoid redundant x-axis labeling. Part A of this figure depicts expression data, whereas part B is an analysis of ATAC-seq data. The stages that were included in the RNA-seq experiment (data from Graf et al, 2014) were different from those profiled by ATAC-seq.

Reviewer #2: Line 332 Any idea if the POU5F1 KO results in less retro-elements to be produced (cow or mice)?

Response: We are not currently aware of any studies that have explored this, but it is an interesting question to pursue.

Reviewer #2: Line 373: I agree that maternal factors may have an initiator role in EGA and oocytes associated with specific conditions in bovine results in a significant increase of embryos achieving development competence and hence passing the 8-cell stage. Does the stored RNAs (like KLF4) in this

high competent group (like OPU after FSH or in vivo early embryos) could indicate which factors would act on the early openings of the 2 cell chromatin?

Response: It is quite possible that high-competence oocytes contain stored RNA corresponding to factors that bind to 2-cell open chromatin. However, because this study did not focus on oocyte competence, we cannot answer this question with the given data, though it would be interesting to pursue in future studies.

Reviewer #3 (Remarks to the Author):

1. Line 155. "The first major transition in chromatin structure mostly involved loss of hyperaccessible sites in 2-cell...". The authors did not examine the zygotes, they cannot know whether the chromatin states in zygotes are the same as that in 2-cell stage. If they want to make this conclusion, they should map the chromatin state of zygotes and demonstrate that the zygotic pattern is similar to 2-cell pattern.

Response: We instead focused our analysis on ATAC-seq peaks that were uniquely observed in GV oocytes. See line 182, beginning with "Over 30,000 ATAC-seq peaks were unique to GV oocytes ..."

Reviewer #3: 2. Line 177. The author conclude that NFkB could play important role in activate major EGA. The conclusion is based on the higher binding frequency in 4- to 8-cell stage than other stage. This is just a correlation. If the author want to make the conclusion, they should knock-down NFkB and check its affect on EGA.

Response: To further support our hypothesis that NFkB activation might be involved in major EGA, we conducted a TF footprinting analysis. This analysis revealed that at the 4-cell stage, NFkB factors may be bound to promoters of genes involved in negative regulation of cell differentiation, including several potential regulators of EGA that were substantially upregulated between the 4- and 8-cell stages (KLF10, KLF17, OTX1, ZSCAN4). See lines 196-200. We were careful to state that NFkB activation "could/may" play an important role in EGA, but that the "contribution of NFkB activity to minor EGA in cattle has not yet been established." The reviewer is correct in stating that this is only a correlation, but it provides an interesting hypothesis to pursue in future studies.

Reviewer #3: Line 192. The author conclude that DUXA play a conserved role during preimplantation development for placental mammals. The conclusion is based on the increased accessibility during bovine development. This is also just a correlation. No causality result has been presented.

Response: The wording of these discussion points has been changed to reflect that we can only hypothesize that DUXA plays a role in major EGA. See line 219: "... the synchronized upregulation of DUXA and increased accessibility of its binding sites during bovine development suggest that DUXA may play a role in bovine EGA. This hypothesis is especially intriguing, considering that DUX family members are implicated in murine and human EGA, and are highly conserved and specific to placental mammals."

Reviewer #3: These are just two examples of the overstatements shown in this paper. Too many similar overstatements can be found throughout the paper.

Response: We revised sections where it appeared that a conclusion was overstated, and included qualifiers, such as 'could', 'may', 'suggests that', etc, to clarify.

Reviewer #3: 3. Blastocyst is an important stage before implantation. The author should check that stage, and compare it with morula stage.

Response: To address this concern, chromatin accessibility was profiled in inner cell mass (3 replicates) as well as embryonic stem cells (3 replicates).

Reviewer #3: 4. The author conclude that maternal and embryonic transcription drive chromatin reorganization. This is the only case showing causality.

Response: We thank the author for recognizing this important mechanistic aspect of our study.

Reviewer #3: 5. Line 313. The author suggest that LINE2 elements could act as enhancers or promoter since these elements express and have accessibility in 8-cell stage. I could not see any relationship between the data and their conclusion. Many similar statements can be found in the transposon section.

Response: These statements have either been revised or removed.

Reviewer #3: Minor comment:

Line 137. "Slate" should be state.

Response: The use of the word 'slate' was intentional. The term "blank slate" refers to something that exists in an original, pristine state. The wording has been rearranged from "blank epigenetic slate" to "epigenetic blank slate" to make it clearer that this expression is being used.

REVIEWER COMMENTS

Reviewer #1 (Remarks to the Author):

The authors addressed most the points that were raised previously and I believe the manuscript is much improved. It represents a very useful dataset and the analysis is done well.

There are two remaining points that I believe are important to address:

1. The authors added an analysis of putative target genes based on TF footprints (Table S4). However, Table S4 does not show a list of the putative gene targets which is what was requested. It shows a summary of “Functional enrichment of genes with promoters marked by open chromatin containing specific TF footprints”, which is interesting but not as informative. It would be helpful to others to make biological sense of the enrichment analysis produced here if the authors could provide a list of all genes that they found associated with all of the motifs identified as enriched at every developmental stage analysed from GV to ICM.

For example, at the 8-cell stage there is enrichment of ZSCAN4 or DUX motifs – what are the chromosome coordinate of the motifs, are these an TSS/intergenic/promotor/etc region and which genes are associated with each motif? Another example is at the morula stage there are CTCF, GATA or KLF motifs – same question as above, what are the coordinates, what is the region and which genes are associated with the motif enrichment? I do not believe this is an unreasonable request and is standard in the field that is provided by other publications. It is data that the authors would have used in this manuscript (to generate the GO term list in Table S4) and it is not very accessible though incredibly useful for the community. Could they please provide this list in an accessible table?

2. While the authors included the ICM stage and bESCs to their dataset it is a pity that they did not include the TE. Could they please acknowledge this absence clearly in the discussion of the paper? This is at the moment not clearly mentioned and a significant shortcoming in my opinion that warrants further investigation in the future by either this group or others.

Reviewer #3 (Remarks to the Author):

The authors revised the manuscript according to my comments. I do not have further comments.

Response to reviewer comments

Reviewer #1 (Remarks to the Author):

The authors addressed most the points that were raised previously and I believe the manuscript is much improved. It represents a very useful dataset and the analysis is done well.

Response: Thanks for the appreciation of our work.

There are two remaining points that I believe are important to address:

1. The authors added an analysis of putative target genes based on TF footprints (Table S4). However, Table S4 does not show a list of the putative gene targets which is what was requested. It shows a summary of “Functional enrichment of genes with promoters marked by open chromatin containing specific TF footprints”, which is interesting but not as informative. It would be helpful to others to make biological sense of the enrichment analysis produced here if the authors could provide a list of all genes that they found associated with all of the motifs identified as enriched at every developmental stage analysed from GV to ICM. For example, at the 8-cell stage there is enrichment of ZSCAN4 or DUX motifs – what are the chromosome coordinate of the motifs, are these an TSS/intergenic/promotor/etc region and which genes are associated with each motif? Another example is at the morula stage there are CTCF, GATA or KLF motifs – same question as above, what are the coordinates, what is the region and which genes are associated with the motif enrichment? I do not believe this is an unreasonable request and is standard in the field that is provided by other publications. It is data that the authors would have used in this manuscript (to generate the GO term list in Table S4) and it is not very accessible though incredibly useful for the community. Could they please provide this list in an accessible table?

Response: To address this concern, we have prepared supplementary datasets which include: 1) Chromosome coordinates of all transcription factor footprints and the closest matching JASPAR motif during each developmental stage and in ESC (Supplementary Datasets 1-7), 2) lists of genes used for each functional enrichment analysis (Supplementary Dataset 8), and 3) An additional supplementary figure showing the genomic distribution of TF footprints during each stage of development relative to annotated genes (Figure S3). These Datasets/Figure are referenced in line 175 of the revised manuscript.

2. While the authors included the ICM stage and bESCs to their dataset it is a pity that they did not include the TE. Could they please acknowledge this absence clearly in the discussion of the paper? This is at the moment not clearly mentioned and a significant shortcoming in my opinion that warrants further investigation in the future by either this group or others.

Response: The absence of open chromatin data for the trophectoderm is now addressed in lines 254-258 of the revised manuscript. We mention that investigation of TE differentiation would be an interesting line of research in the future.

Reviewer #3 (Remarks to the Author):

The authors revised the manuscript according to my comments. I do not have further comments.

Response: Thanks for your efforts helping improve the original submission.